# The Extreme Environment Microbiome Catalog (EEMC): a global resource for microbial diversity and antimicrobial discovery

Microorganisms in extreme environments represent a promising source of novel metabolites, yet their global diversity and biosynthetic potential remain underexplored. Here, we reconstruct 78,213 bacterial and archaeal genomes from 2293 publicly available metagenomes and 3214 microbial isolates to establish a unified database, the Extreme Environment Microbiome Catalog (EEMC). The EEMC expands known global phylogenetic diversity, encompassing 32,715 representative species and nearly 4 billion non-redundant genes, 63.00% and 19.21% of which are previously unannotated, respectively. It also comprises 163,693 biosynthetic gene clusters, grouped into 64,733 gene cluster families, 58.68% of which are classified as novel, underscoring the functional diversity of microbial communities across various extreme habitats. We further develop protein large language models to predict genome-encoded candidate antimicrobial peptides (cAMPs) from the EEMC, identifying 3032 non-toxic candidates. Of 100 synthesized peptides, 84% demonstrate antibacterial activity, and all 50 tested cAMPs exhibit low cytotoxicity. Notably, six of the most potent cAMPs show significant efficacy against multidrug-resistant, Gram-negative pathogens in vitro, indicating their biomedical potential. Together, our study establishes the EEMC as a foundational resource for uncovering novel microbial lineages and biosynthetic capabilities, highlighting its substantial potential for drug discovery and laying the foundation for future advances in biotechnology and biomedicine.

Extreme environments, defined by harsh conditions such as extreme pH, salinity, temperature, and pressure, host microorganisms with broad phylogenetic and functional diversity spanning all domains of life[1,2]. These microbes have evolved distinct metabolic strategies and physiological adaptations to survive and thrive under such stresses[3,4], offering a valuable source of novel lineages[5,6], enzymes[7,8], and bioactive metabolites[9–11]. These molecules not only underpin key ecosystem functions, such as biogeochemical cycling and interspecies interactions, but also hold potential for biotechnological and biomedical applications[1,2]. Nevertheless, current knowledge of microbial functions and metabolites is heavily skewed toward cultured lineages, leaving much of the metabolic capacities of uncultivated microbes unexplored[12–14].

Recent advances in next-generation sequencing technologies and computational tools have enabled genome recovery from metagenomes or single cells without cultivation, providing genome-scale insights into the taxonomic diversity, metabolism and ecological roles of microbes[15–17]. Indeed, several small-scale studies have demonstrated

✉ e-mail: yuezhen@genomics.cn; xue.1@dlut.edu.cn; yinpeng@genomics.cn; chenhaixin@genomics.cn

the utility of these approaches in expanding phylogenomic representation and uncovering functional diversity in microbiomes from extreme environments, such as glaciers and acid mine drainage[18–20]. However, most existing studies remain geographically restricted and habitat-specific, lacking large-scale, integrative analyses across diverse extreme habitats at a global scale. This gap limits our understanding of the distribution, functions and ecological relevance of microbial communities and biosynthetic capacities of global extreme environment microbiomes, thereby hindering the discovery and development of novel enzymes and bioactive compounds from these untapped microbial reservoirs.

Antibiotic resistance is a major global health threat, with an estimated 39.1 million deaths from 2025 to 2050[21]. Despite the urgency, the development of new antibiotics has stagnated, with no new classes of antibiotics discovered since the 1980s due to escalating costs and waning investment[22,23]. Therefore, innovative approaches for antibiotic discovery are urgently needed.

Microbial natural products, also known as secondary metabolites, serve as a rich source of antimicrobial compounds[24–26]. Among these, antimicrobial peptides (AMPs), particularly ribosomally synthesized and posttranslationally modified peptides (RiPPs), are of special interest due to their membrane-disrupting mechanisms and reduced susceptibility to resistance development[27]. The genetically encoded nature of RiPPs enables genome-guided mining of their products, and several studies on human gut and marine microbiomes have successfully predicted RiPPs with antimicrobial activity[28–31]. Although RiPP-type biosynthetic gene clusters (BGCs) from extreme environments have shown considerable biosynthetic novelty, their functional potential, particularly therapeutic relevance, remains underexplored, with limited experimental validation beyond in silico predictions[18,20].

Recently, artificial intelligence (AI) has enabled significant progress in AMP discovery, including AMP prediction[30,32–34], optimization, and de novo design[35,36]. While most models employ traditional machine learning (e.g., iAMPred[33] and amPEP[34]) or deep learning algorithms (e.g., Macrel[37] and AMPScanner[32]) for AMP prediction, their reliance on relatively small and potentially biased datasets often limits robustness and generalizability. Although several recent deep learning models, such as the human cAMP model[30] and APEX[38,39], demonstrate improved generalization, they are still trained primarily on curated AMP and non-AMP peptide datasets rather than on large-scale protein representations, and therefore do not leverage the broad sequence-function priors captured by protein language models. Moreover, most studies prioritize AMP activities while neglecting toxicity, which remains a major cause of clinical failure[40]. Concurrently, protein large language models (pLLMs), such as ProtT5[41], ESM2[42], and ESM3[43], have emerged as powerful deep learning tools that leverage large-scale protein sequence datasets to capture biological properties related to structure and function. These advances have supported a range of protein function prediction tasks, including classification, interaction prediction, and structure-function mapping. Given these capabilities, applying pLLMs to systematically predict both the antimicrobial activity and toxicity of RiPP peptides derived from extreme environment microbiomes represents a promising and necessary endeavor.

Here, to explore the taxonomic structure and biosynthetic potential of extreme environment microbiomes at a global scale, we first analyzed 2293 publicly available metagenomes and 3131 reference isolate genomes (REFs) from public databases, complemented by 83 in-house isolate genomes, to establish a unified resource termed the Extreme Environment Microbiome Catalog (EEMC). The EEMC comprises 78,213 bacterial and archaeal genomes, nearly 4 billion non-redundant gene clusters, and 163,693 BGCs, highlighting its considerable taxonomic and biosynthetic diversity and novelty. Furthermore, building upon the EEMC, we developed pLLM-based models to identify candidate non-toxic AMPs (cAMPs) from RiPP-type BGCs, yielding a total of 3032 cAMPs. Among 100 synthesized cAMPs, 84% showed antibacterial activity, with all 50 tested cAMPs showed low toxicity to mammalian cells. Mechanistic analysis of selected cAMPs indicated membrane-disrupting activity. Together, the EEMC provides a comprehensive resource for exploring microbial diversity and biosynthetic capacity in extreme environments, with significant implications for drug discovery and broader biotechnological applications.

## Results

### The EEMC includes 20,610 previously uncharacterized microbial species

To systematically explore the taxonomic diversity and biosynthetic potential of microorganisms inhabiting global extreme environments, we first sought to establish a genome-resolved data resource as the foundation for downstream analyses (Supplementary Fig. 1). To this end, we collected metagenomic sequencing data from 2293 publicly available metagenomes covering a broad range of defined extreme environments, including deep-sea, cryosphere, subsurface systems, hypersaline regions, hyperacid regions, terrestrial geothermal regions, and hyperarid regions (Fig. 1a, Supplementary Fig. 1, Supplementary Data 1). We performed assembly and binning of 2293 metagenomes, and recovered a total of 74,999 metagenome-assembled genomes (MAGs) (Fig. 1b). Furthermore, to avoid the loss of information from culturable strains, we collected 3214 isolate genomes, including 83 in-house isolate genomes derived from cold seeps and 3131 REFs collected from the NCBI RefSeq genome database[44] (Fig. 1b). We reconstructed a total of 78,213 genomes to generate the genome component of the EEMC catalog (Fig. 1b, Supplementary Data 2). All of the genomes met or exceeded the medium-quality level of the Minimum Information about a Metagenome-Assembled Genome (MIMAG) criteria (completeness ≥ 50% and contamination < 10%) (Fig. 1b)[45] with a mean completeness rate of 76.84% ( ± 15.29%) and a mean contamination rate of 2.75% ( ± 2.31%) (Fig. 1c). Among these genomes, 57,946 exhibited a quality score (QS) above 50 (calculated as completeness - (5 × contamination)), 18,988 exhibited a completeness greater than 90%, contamination less than 5%, and 9030 were assigned as high quality featuring the presence of the 23S, 16S and 5S rRNA genes and at least 18 tRNAs (Fig. 1b, Supplementary Data 2).

We next set out to explore the taxonomic components and novelty of the EEMC. Using an average nucleotide identity (ANI) threshold of 95% and an alignment fraction (AF) threshold of 30%[18], the 78,213 genomes were clustered into 32,715 representative operational taxonomic units (OTUs), of which 93.33% and 5.94% were represented by MAGs and isolate genomes exclusively, respectively, whereas 0.73% of the OTUs were represented by both MAGs and isolate genomes, indicating that MAGs and isolate genomes were highly complementary (Supplementary Data 2). Comparisons with published catalogs indicated the novelty of the 32,715 species-level OTUs, with 99.85%, 95.56%, and 88.42% showing low sequence identity to the 957 TARA genomes[46], 22,732 GEM genomes[47] and 24,195 GOMC genomes[19], respectively (Fig. 1d, Supplementary Data 3). A total of 28,261 (86.39%) OTUs did not map to any of these reference genomes. The substantial novelty of the genomes suggests that EEMC provides a unique microbial and metabolic composition compared with other ecosystems.

We then annotated the representative OTUs to identify their taxonomic diversity using the Genome Taxonomy Database (GTDB) Toolkit. The OTUs represented 190 phyla, 506 classes, 1446 orders, 3266 families, and 8397 genera (Supplementary Data 2). Among these phyla, the most highly represented were Pseudomonadota (n = 7104, 21.71%), Bacteroidota (n = 3015, 9.22%), Actinomycetota (n = 2560, 7.83%) and Patescibacteria (n = 2164, 6.61%). Of the OTUs identified, 8 (0.02%) phyla, 47 (0.14%) classes, 254 (0.78%) orders, 987 (3.02%) families, 5274 (16.12%) genera, and 20,610 (63.00%) species were not assigned annotations in the GTDB r220, indicating potential novel lineages (Supplementary Data 2). Among these 20,610 putative novel

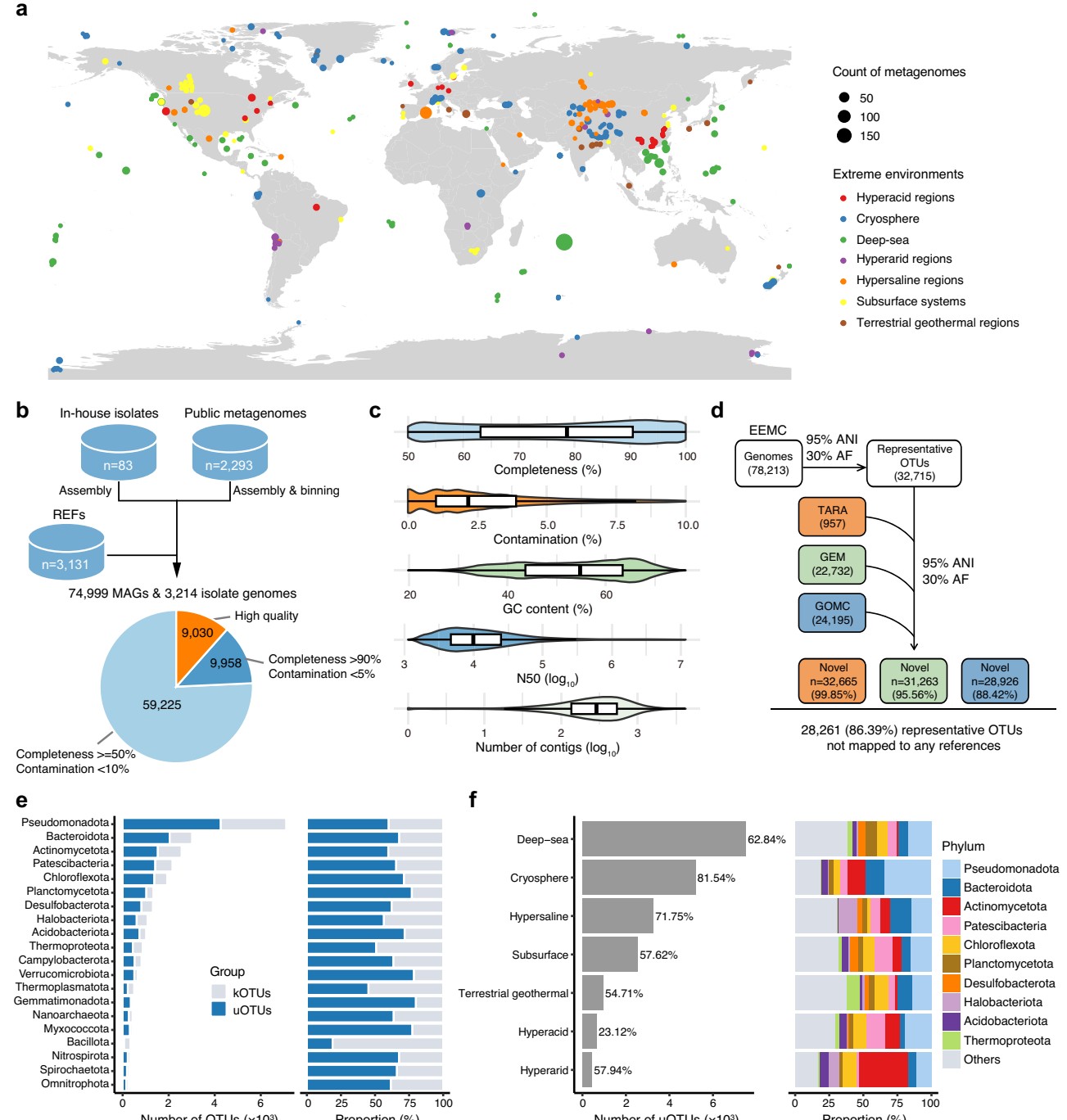

**Fig. 1 | The environmental distribution of EEMC genomes and their species-level clustering. a** Geographic distribution of 2293 publicly available metagenomes within each habitat. **b** A total of 78,213 genomes were recovered from 2293 metagenomes and 3214 cultivated isolates. All genomes met the quality of completeness ≥ 50% and contamination < 10%. **c** Distribution of quality metrics for the genomes ($n = 78,213$). Boxes represent the interquartile range between the first and third quartiles, with the line inside indicating the median value. Violin plots show the full distribution of the data, including the minimum and maximum values. **d** EEMC genomes were clustered into 32,715 species-level OTUs, based on 95% ANI

and 30% AF. Representative genomes from TARA ($n = 957$), GEM ($n = 22,732$), GOMC ($n = 24,195$) were then included in the clustering to evaluate the novelty of the EEMC species. **e** The number and proportion of kOTUs and uOTUs across the top 20 phyla. **f** The number (left panel) and top 10 phyla composition (right panel) of uOTUs across extreme environments. The proportion of uOTUs in each environment is labeled to the right of each bar in the left panel. OTU operational taxonomic unit, ANI average nucleotide identity, AF alignment fraction, kOTU known OTU, uOTU unclassified OTU.

species, 13,785 had a QS greater than 50, and 1483 met the high-quality genome criteria (Supplementary Data 2). Notably, the OTUs composed of MAGs covered all 190 phyla, while the OTUs composed of isolate genomes alone and those composed of both MAGs and isolate genomes covered 32 and 23 known phyla, respectively, highlighting the

high novelty of MAGs. Moreover, the majority of unclassified OTUs (uOTUs) were singleton genomes (67.62%), markedly exceeding the proportion observed in known OTUs (kOTUs) (54.41%) (Supplementary Fig. 2a). We further found that the proportion of uOTUs within the dominant phyla (≤ 80%), such as Gemmatimonadota and

Planctomycetota, was lower than that in rare phyla ( > 85%), including Bdellovibrionota and Hydrogenedentota (Fig. 1e, Supplementary Fig. 2b). This pattern indicates the critical contribution of the EEMC in recovering rare species of extreme environment microbiomes. Overall, the EEMC expands the previous GTDB database by 18.22% (20,610 novel genomes compared to 113,104 references), significantly increasing the diversity of global microbiomes.

Next, we investigated the distribution of the OTUs across various extreme habitats. We found that 81.54% (5,205/6,383), 71.75% (3,268/4,555), and 62.84% (7,516/11,961) of OTUs recovered from cryosphere, hypersaline regions, and deep-sea respectively were uOTUs (Fig. 1f). Distinct phylum-level patterns of uOTUs were observed across extreme environments. For instance, Pseudomonadota, Bacteroidota, and Actinomycetota were highly represented among uOTUs from the cryosphere, while Halobacteriota were frequent among those from hypersaline regions. Actinomycetota contributed substantially to uOTUs in hyperarid regions, and Chloroflexota and Thermoproteota were notable among uOTUs from terrestrial geothermal habitats (Fig. 1f). Overall, these observations highlight the broad phylogenetic diversity of uOTUs captured by the EEMC across extreme environments.

## The EEMC comprises nearly 4 billion non-redundant genes, revealing extensive diversity and notable novelty

To investigate the functional landscape of the EEMC, we predicted 5,215,328,758 open reading frames (ORFs), from the original unbinned contigs and subsequently clustered them into 3,999,744,836 non-redundant gene clusters using thresholds of 80% coverage and 95% identity; the representative genes of these clusters were designated as unigenes. The number of unigenes increased steadily with increasing sampling or sequencing depth and showed no signs of plateauing (Supplementary Fig. 3a, b). Notably, among the extreme environments, the deep-sea contributed both the largest number of samples and unigenes; moreover, the number of unigenes increased most rapidly with sampling depth in this environment. These findings suggest the substantial genetic diversity of extreme environments, particularly the deep-sea, indicating that further sampling and sequencing are needed to explore this diversity fully.

Taxonomic and functional annotations of the non-redundant gene set were performed using the NR, UniRef50, and Swiss-Prot databases[48,49], with 77.74%, 69.24%, and 27.83% of the unigenes annotated in the corresponding databases. Functional annotations were also performed using the COG, KEGG, CAZy, GO, CARD, and VFDB databases[50–55], producing annotations for 44.36%, 28.77%, 9.72%, 4.08%, 3.74%, and 1.49% of the unigenes, respectively (Supplementary Fig. 3c). A total of 784,252,225 (19.61%) unigenes were not annotated in any of the above databases and were thus considered novel genes. Among the extreme environments, the deep-sea environment harbored the largest number of novel genes (408,199,569), while the hyperacid environment exhibited the highest proportion of novel genes (23.0%) (Fig. 2a). In all the extreme environments, a predominant proportion of the annotated genes were related to DNA repair (e.g., K03701, K00525, K03657), material transport (e.g., K02004, K01990, K06147) or metabolic regulation (e.g., K00626, K12132) (Supplementary Fig. 3d, Supplementary Data 4), indicating the critical importance of robust DNA maintenance, versatile transport, and adaptive metabolic and regulatory mechanisms for microbial survival under harsh conditions. Moreover, we observed the widespread presence of pathways related to the biosynthesis of antibiotics and secondary metabolites (ko01130, ko01110) (Supplementary Fig. 3d, Supplementary Data 4), suggesting that microorganisms from extreme environments likely employ chemical strategies for competition and interaction, an ecological mechanism broadly observed across diverse microbial ecosystems[2,11,56]. Given the considerable number of novel genes and biosynthetic features uncovered in the

EEMC, the microbes in these habitats may therefore represent valuable reservoirs of novel natural products. Notably, the number of unigenes shared across all environments was remarkably limited (n = 147); conversely, approximately 99.79% of the total unigenes were specific to a single environment (Fig. 2b). As expected, unigenes shared across all environments included housekeeping genes, such as ribosome-associated core genes (e.g., GGM92914.1, WP_003863308.1, WP_003175898.1), as well as genes linked to adaptation to extreme conditions, including those involved in DNA stability (e.g., GBG13987.1, WP_001124935.1) and others that enhance microbial tolerance to harsh environments (e.g., WP_017980795.1, WP_084592356.1, WP_000429838.1) (Supplementary Data 5).

To further evaluate the novelty of our gene resources, we predicted 203,220,315 ORFs directly from the 78,213 microbial genomes, and compared them with genome-derived gene datasets from GEM and GOMC. Genes from the three datasets were clustered into 116,034,223 gene clusters, of which 76,824,985 originated from our EEMC gene catalog (Fig. 2c; at thresholds of 70% coverage and 80% average amino acid identity (AAI)). Among these, only 15.68% (12,045,620/76,824,985) clusters were shared with either the GEM or GOMC datasets. Consistent with the novelty distribution of the unigenes, clustering against the GEM and GOMC datasets revealed that the deep-sea environment harbored the largest number of novel gene clusters (27,177,489), whereas the hyperacid regions exhibited the highest proportion of novel clusters (88.16%) (Fig. 2d). We next examined the functional characteristics of genes from the EEMC by performing comparative analyses using the VFDB, CARD, and CAZy databases to identify potential virulence factors (VFs), antibiotic resistance genes (ARGs), and carbohydrate-active enzymes (CAZymes), respectively. For each category, genes identified from the EEMC gene catalog were clustered together with those from the GEM and GOMC gene catalogs using the same criteria (70% coverage and 80% AAI). Specifically, 4,340,855 potential VFs were identified from the EEMC gene catalog, resulting in 1,218,145 VF clusters, of which only 21.48% (261,641/1,218,145) were shared with either the GEM or GOMC datasets (Fig. 2e). Similarly, 269,235 ARG clusters and 3,513,158 CAZyme clusters were obtained from the EEMC gene catalog, with only 19.65% (52,903/269,235) and 17.76% (623,818/3,513,158) of them, respectively, overlapping with the GEM or GOMC datasets (Fig. 2f, g). All these together highlight the distinctiveness of the EEMC gene catalog.

## The EEMC harbors broad and diverse biosynthetic potential, comprising over 163,000 biosynthetic gene clusters

To explore the potential application of EEMC in drug discovery, we next investigated its biosynthetic capacity for secondary metabolites encoded by BGCs. We first used antiSMASH (version 7.0.1) on the 78,213 microbial genomes to identify 163,693 BGCs, of which 143,958 (87.94%) originated from genomes with QS > 50. As expected, the proportion of complete BGCs was lower in genomes with QS ≤ 50 (2.90%) than in the full genome set (19.97%) and higher-quality genomes (QS > 50) (22.31%) (Supplementary Fig. 5c). In addition, only 14.63% of all partial BGCs (19,161 out of 131,001) came from genomes with QS ≤ 50 (Supplementary Fig. 5c), indicating that fragmented BGCs are not predominantly contributed by lower-QS genomes. The 163,693 BGCs were further categorized into eight classes, with lengths ranging from 4691 to 288,454 bp (Supplementary Data 6). The distribution of class-level composition remained consistent across the full and QS-stratified genome datasets (Supplementary Fig. 5d). These results indicate that the inclusion of a limited number of lower-quality genomes does not materially affect our conclusions regarding overall BGC diversity and distribution patterns.

Among the eight BGC classes, the RiPPs encoded by 165 phyla served as the most abundant class of BGCs, accounting for 28.60% (n = 46,816) of the total BGCs (Supplementary Fig. 5a). We also

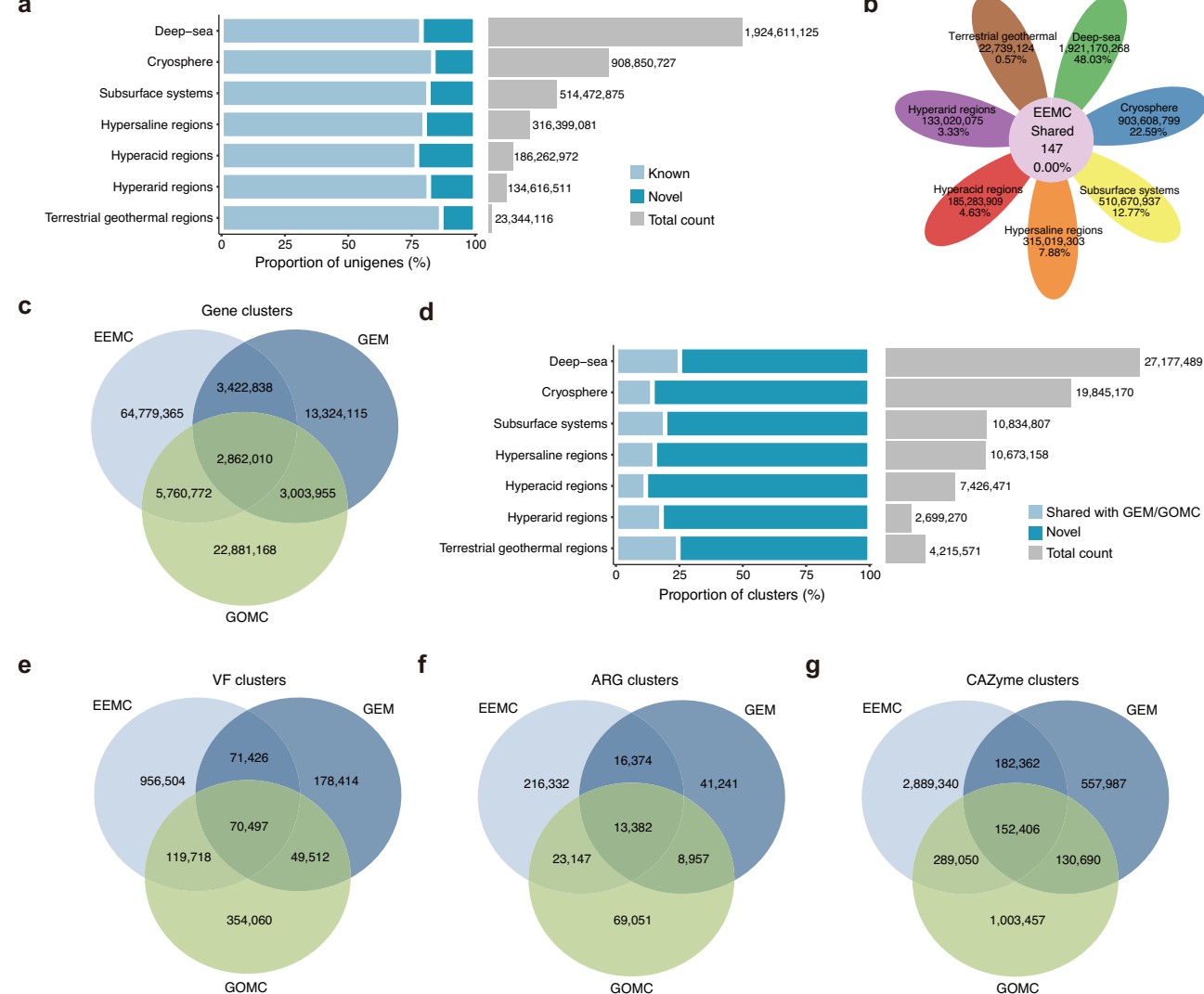

**Fig. 2 | Overview of non-redundant genes and genome-derived gene clusters in the EEMC. a** Novelty and total number of unigenes identified in each extreme environment. **b** Number of unigenes unique to each extreme environment and those shared across all environments. **c** Venn diagram shows the overlap of genome-derived gene clusters among the EEMC, GEM, and GOMC datasets. The intersections represent shared clusters, while the non-overlapping areas indicate unique clusters specific to each dataset. **d** Novelty and total number of genome-derived gene clusters identified in each extreme environment. **e–g** Venn diagrams show the overlap of genome-derived gene clusters associated with VFs (**e**), ARGs (**f**), and CAZymes (**g**) among the EEMC, GEM, and GOMC datasets. VF virulence factor, ARG antibiotic resistance gene, CAZyme carbohydrate-active enzyme.

identified 40,616 (24.81%) terpene clusters from 104 phyla, 20,608 (12.59%) non-ribosomal peptide synthetase (NRPS) clusters from 96 phyla, 3667 (2.24%) polyketide synthase (PKSI) clusters from 57 phyla, 3225 (1.97%) PKS-NRPS hybrid gene clusters from 46 phyla, and 107 (0.06%) saccharide clusters from 10 phyla (Fig. 3a, Supplementary Fig. 4&5a). More than half of the BGCs (n = 90,548, 55.32%) were identified in phyla Pseudomonadota (n = 58,477, 35.72%), Actinomycetota (n = 19,981, 12.21%), and Bacteroidota (n = 12,090, 7.39%) (Supplementary Fig. 5b, Supplementary Data 7). Among the top 15 phyla, members of the phyla Acidobacteriota and Myxococcota had a high BGC density, with up to 110 and 84 found in the genomes, respectively (Fig. 3a, Supplementary Data 6&7), aligning with those from the GEM catalog previously reported[47].

We further investigated the biosynthetic potential of each extreme environment. Among the identified BGCs, RiPP-type BGCs were dominant in most extreme habitats, whereas terpene-type BGCs were dominant in the cryoshpere (Fig. 3b, Supplementary Data 8), consistent with the previously reported prevalence of terpenes in the glacier microbiomes[18]. Moreover, genomes from deep-sea and cryosphere had the highest biosynthetic potential, with 53,587 (32.74%) and 40,363 (24.66%) BGCs identified, respectively (Fig. 3b). In contrast, the genomes from terrestrial geothermal regions and hyperarid regions had the lowest biosynthetic potential, with 34.12% and 55.88% of BGCs identified in isolate genomes, respectively (Fig. 3b, Supplementary Data 8). However, as these habitats also featured the fewest assembled genomes, more metagenomic analyses are needed to provide more comprehensive and deeper insight into their biosynthetic potential.

To assess the degree of novelty of the biosynthetic potential within the EEMC, we clustered the 163,693 BGCs into 64,733 non-redundant gene cluster families (GCFs) and 2178 gene cluster clans (GCCs) (Supplementary Data 6, Methods). Subsequently, we compared the GCFs and GCCs with the reference BGCs of the BiG-SLiCE database, assigning 37,984 (58.68%) GCFs and 1528 (70.16%) GCCs as novel (Supplementary Data 9). Most BGC types had more than 50% novel GCFs or GCCs, including terpene, NRPS, PKSI, PKS-NRP hybrids, and saccharides (Fig. 3c, Supplementary Data 9). Additionally, the majority of the novel diversity (24,437 GCFs, that is 64.33%)

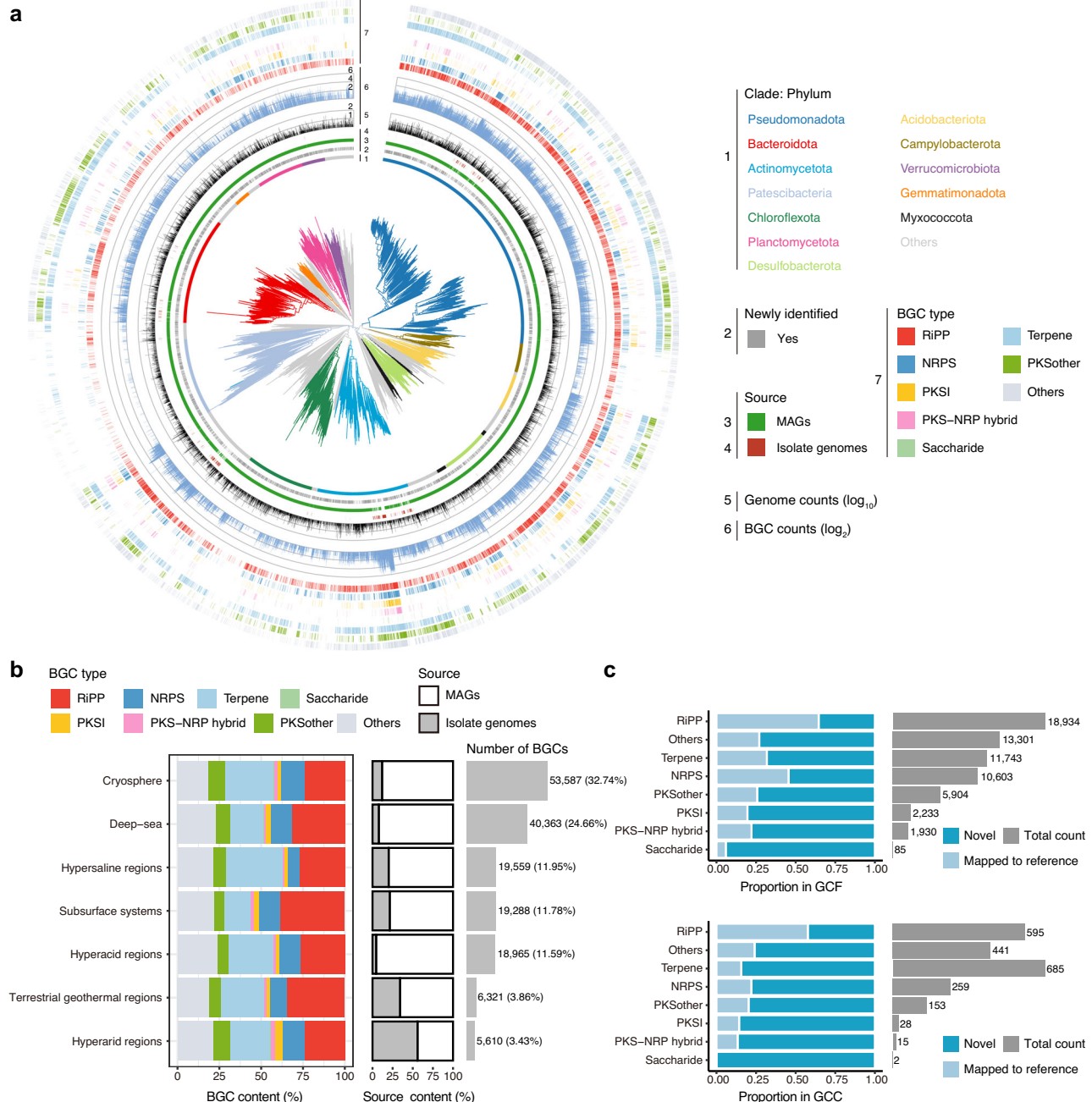

**Fig. 3 | Novelty and phylogenomic distribution of the EEMC genomes and their biosynthetic potential. a** A bacterial phylogenetic tree was built for 32,715 representative OTUs on the basis of a concatenated alignment of 120 universally distributed bacterial single-copy genes. The color of the branches and the nearest outer layer (1) indicate the phylum of the species. The other outer layers indicate (for each species) the novelty compared to the GTDB r220 database (2), the genome source (3, 4), the genome count (5), the largest number of BGCs among its genomes (6), and the presence of each BGC type among its genomes (7). **b** The left panel shows the content of each BGC type across seven extreme environments. The middle panel shows the percentage of BGCs identified in MAGs and isolate genomes. The right panel shows the count of identified BGCs in each extreme environment. **c** The 163,693 BGCs were clustered into 64,733 GCFs and 2178 GCCs, with their novelty and the total count across different types indicated. OTU operational taxonomic unit, BGC biosynthetic gene cluster, MAG metagenome-assembled genome, GCF gene cluster family, GCC gene cluster clan.

corresponded to predicted terpenes ($n = 8009$), RiPPs ($n = 6719$), or other natural products ($n=9709$) (Supplementary Data 9), whose BGC structures are denser and thus less likely to be fragmented during metagenomic assembly than are the NRPS and PKS clusters[15]. The overall composition of BGC classes and the novelty proportions within each BGC class were largely consistent across the full BGCs and the two completeness-stratified subsets (Fig. 3c, Supplementary Fig. 5d, e), indicating that cluster fragmentation had only a modest impact on

distance-based novelty scoring, and that the high proportion of novel BGCs in the EEMC likely reflects genuine biosynthetic divergence rather than assembly artifacts.

Considering that GCCs are likely to represent highly diverse biosynthetic functions, we further analyzed the data at the GCF level, which aims to provide a more fine-grained grouping of BGCs that are predicted to encode similar natural products[15]. The biosynthetic diversity also varied across the prokaryotic domains. We identified a

total of 2480 archaeal GCFs, of which nearly all ($n = 2473$, 99.72%) being archaeal-specific, mainly from the phyla Halobacteriota, Thermoproteota, and Thermoplasmatota (Supplementary Data 6, 10). Among the bacterial phyla, Pseudomonadota, Actinomycetota, and Bacteroidota had abundant GCFs (Supplementary Fig. 5f, Supplementary Data 10). Furthermore, to assess how biosynthetic diversity scales with genomic sampling, we performed rarefaction analyses at the GCF level. We observed the phyla Pseudomonadota and Actinomycetota (Supplementary Fig. 5f), as well as the genera *JAKAGC01* and *Streptomyces* had the highest potential of producing secondary metabolites with accumulation curves that remained unsaturated (Supplementary Fig. 5g), consistent with previous findings[19,57]. These findings confirm that the EEMC catalog harbors substantial biosynthetic potential, alongside the observed taxonomic diversity, thereby largely expanding the current global BGC repository.

RiPPs are considered a promising source of AMPs due to their remarkable chemical and functional diversity. Therefore, we next sought to investigate the diversity of the RiPP clusters to expand our understanding of their biosynthetic capabilities. A total of 46,816 RiPP-type BGCs were clustered into 18,934 GCFs and 595 GCCs, with 35.49% ($n = 6,719$) and 42.35% ($n = 252$) assigned as novel, respectively (Fig. 3c). We found that RiPP families were observed predominantly in a single extreme habitat, with 17,830 (94.17%) GCFs detected in only one extreme habitat (Supplementary Fig. 6a). Similar environment-associated patterns of biosynthetic diversity have also been observed in other large-scale microbiome studies, such as fermented food and human microbiomes[58,59].

We then analyzed the RiPP diversity and capability across phylogenetic trees. The phyla Pseudomonadota, Actinomycetota, Desulfobacterota, and Bacteroidota had the highest GCF diversity of RiPPs, with more than 99% of their GCFs being phylum-specific (Supplementary Data 11), implying that the biosynthetic potential of certain RiPPs may be restricted to specific taxa. We defined RiPP capacity as the count and density of RiPP clusters (the number of RiPP clusters per megabase pair of the genome) within the genome. Our findings revealed that the phylum Actinomycetota was the most RiPP-rich phyla, with up to 18 RiPP clusters identified within the isolate genomes of *Crossiella sp023094165*, followed by the phyla Pseudomonadota, Chloroflexota, Myxococcota, and Halobacteriota (Supplementary Data 6, Supplementary Fig. 6b). Additionally, dense RiPP distributions were observed in the phyla Thermoproteota and Thermoplasmatota, both of which were archaeal MAGs recovered from hydrothermal vents (Supplementary Fig. 6c). Most genomes with a high RiPP density ($\geq 2$ RiPPs/Mb) were MAGs ($n = 165$, 94.83%), probably owing to the short lengths of metagenomic assemblies (Supplementary Data 12). These observations suggest that RiPPs may be linked to ecological adaptations and host interactions[28], though sampling and sequencing biases may also play a role[60].

## Deep learning models for predicting antimicrobial activity and toxicity

To discover novel and non-toxic AMPs, we systematically analyzed 46,816 RiPP-type BGCs derived from the EEMC genomes using a two-step AI-assisted strategy. First, core peptides and their corresponding classes were predicted using DeepRiPP, a machine learning tool capable of identifying RiPP core peptides independently of genomic context[29]. Then, pLLM-based classifiers were constructed to predict non-toxic AMPs from the predicted core peptides followed by experimental validation, demonstrating the promise of EEMC as a rich source for AMP discovery (Supplementary Fig. 1).

First, ORFs of 46,816 RiPP clusters were predicted by prodigal and submitted to DeepRiPP, leading to the discovery of 11,379 non-redundant core peptides, which were mainly lassopeptides ($n = 7414$, 65.16%) and class-II-lantipeptides ($n = 1737$, 15.26%) (Supplementary Fig. 7a). We observed that several types of RiPPs, such as auto-

inducing peptides, microviridin, linear azole-containing peptides, and cyanobactin, were enriched in specified environments. Next, to train and validate models for identifying non-toxic AMPs, we compiled a comprehensive dataset by integrating peptide sequences from multiple studies, including 25,004 AMPs, 30,205 non-AMPs, 6318 toxic peptides and 6643 non-toxic peptides (Fig. 4a). Then, we created the Metagenomics-AI (MAI) framework, which incorporates several pLLMs and a classification head consisting of one or more fully linear layers with ReLU activation. These models were trained to predict antimicrobial activity and mammalian cytotoxicity of the predicted core peptides. Specifically, we adopted ESM2 (650M and 3B parameter versions), ESM3, and PTrans models, due to their proven effectiveness in capturing sequence-level representations and supporting various downstream protein prediction tasks[41–43]. Moreover, in light of evidence that certain AMPs exhibit selective activity against Gram-positive (G+) or Gram-negative (G−) bacteria[61,62], we constructed separate classification models tailored to predict (i) anti-G+ activity, (ii) anti-G- activity, (iii) general antimicrobial activity (Supplementary Data 13), and (iv) cytotoxicity against mammalian cells. Accordingly, four labeled datasets corresponding to these four properties were curated and processed using the above pLLM-based models.

We observed that the model performance on these datasets was strongly correlated with the number of parameters of the pLLMs (Fig. 4b, Supplementary Fig. 8a, b). The ESM2-3B emerged as the top performer, with an accuracy of 94.56%, 96.87%, 96.67% and 84.66% in predicting antimicrobial activity, anti-G- activity, anti-G+ activity and toxicity, followed by ESM3 and PTrans (Fig. 4b). The ESM2-650M yielded the lowest scores across all datasets, except for the AMP global dataset. We then compared the performance of pLLM-based models with the popular AMP prediction tools and peptide toxicity model, including Macrel[37], AMPScanner[32], iAMPpred[33], amPEP[34], cAMP model[30], and ToxinPred[63]. The ESM2-3B-based models achieved more than 10% improvement in performance over the known AMP models in terms of accuracy, F1-score, area under the receiver-operating characteristic curve (AUC), and Matthews correlation coefficient (MCC) (Supplementary Fig. 8c, Supplementary Data 14). However, ToxinPred had a higher accuracy, F1-score, AUC, and MCC in predicting toxicity than the pLLM-based models, probably due to the class imbalance of the test dataset (Supplementary Fig. 8a–b, Supplementary Data 14). Together, these results indicate that the ESM2-3B-based, ESM3-based and PTrans-based models outperformed the known models; therefore, these models were selected for predicting cAMPs.

Using these models, we screened 11,379 core peptides to identify AMPs with high confidence for further experimental validation. Briefly, each peptide was assigned classification and confidence scores by each pLLM-based model. Non-toxic AMPs predicted by models that were based on the same pLLMs were selected and ranked according to their scores (Supplementary Fig. 9a–g, Supplementary Data 15). The top 50% of peptides were considered cAMPs (Fig. 4c–e, Supplementary Data 15). We thus obtained a total of 3032 cAMPs, of which 2702, 1938, and 1729 were identified from AMP global models (Fig. 4c), anti-G+ models (Fig. 4d), and anti-G- models (Fig. 4e), respectively. Among the 3032 cAMPs predicted, 108 were classified as high-confidence cAMPs by all models. To confirm that these predictions were made on novel sequences, we performed a pairwise Needleman-Wunsch similarity analysis between the EEMC-derived core peptides and the training datasets. Most RiPP sequences shared low similarity with the global AMP and toxicity training datasets (Supplementary Fig. 7b, c, Supplementary Data 15), indicating that the cAMP predictions were applied to sequences independent of the training datasets.

We further randomly selected 54 peptides from these high-confidence cAMPs and 54 from the remaining cAMPs, resulting in a total of 108 cAMPs for experimental testing. Of the 108 cAMPs, 88 cAMPs were predicted from MAGs, and 20 were predicted from

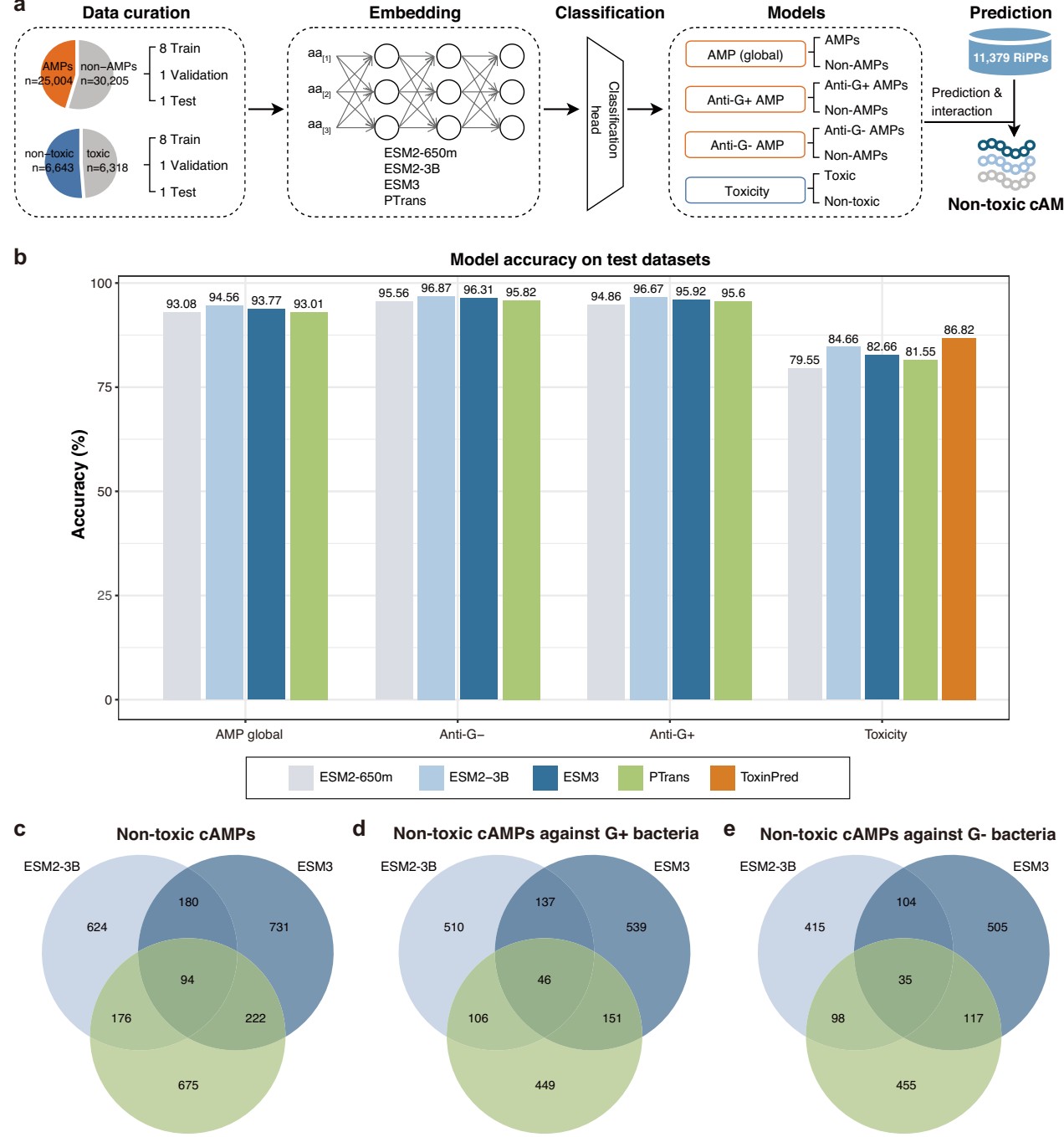

**Fig. 4 | Overview and performance of deep learning models used for predicting AMPs and toxicity. a** Data collection and deep learning architecture of the pLLM-based models used for predicting antimicrobial activity and toxicity. The AMP global models were trained on 25,004 AMP sequences and 30,205 non-AMP protein sequences. The AMPs with anti-G+ or anti-G- label and non-AMPs were further used to train anti-G+ AMP models and anti-G- AMP models, respectively. The toxicity models were trained on 6318 toxic and 6643 non-toxic protein sequences. The trained pLLM-based models were used to predict the antimicrobial activity and toxicity of 11,379 RiPPs. **b** Accuracy of models based on ESM2-650M, ESM2-3B,

ESM3, and PTrans on test datasets with binary labels (Global AMP, anti-G+, anti-G- and Toxicity). Members of the global AMP dataset with proven activity against at least one microbial organism were considered active. **c–e** Venn diagrams show the overlap of cAMPs against microbes (**c**), Gram-positive bacteria (**d**) and Gram-negative bacteria (**e**) identified by the ESM2-3B-, ESM3- and PTrans-based models. AMP antimicrobial peptide, pLLM protein large language model, G+ Gram-positive bacteria, G- Gram-negative bacteria, RiPP ribosomally synthesized and post-translationally modified peptide, cAMP candidate non-toxic antimicrobial peptide.

isolates (Supplementary Data 15). All 108 selected linear cAMPs were shorter than 30 amino acids and lacked post-translational modifications (PTMs), due to the limited reliability of PTM prediction[29,64] and length constraints in chemical synthesis. Eight of the peptides failed

after three attempts of chemical synthesis and the remaining 100 cAMPs were subjected to further validation. The 100 cAMPs were mainly derived from lasso peptide class (87/100) and class II lanti-peptide (9/100) BGCs, predominantly from Actinomycetota (31

cAMPs), Pseudomonadota (24 cAMPs), and Acidobacteriota (8 cAMPs) (Supplementary Data 16). In terms of habitat of origin, 31 cAMPs were identified from cryosphere, followed by 17, 17 and 13 cAMPs from subsurface systems, hypersaline regions, and hyperacid regions, respectively (Supplementary Data 16), highlighting a variety of extreme environment microbiomes as rich sources for novel AMPs.

## cAMPs from the EEMC with efficacy against a range of pathogens

To evaluate and characterize antimicrobial activity of cAMPs, we tested the 100 cAMPs against a panel of 11 bacterial strains associated with human health at a concentration of 60 $\mu$M in liquid media, including *Enterococcus faecalis* American Type Culture Collection (ATCC) 29212, *Staphylococcus aureus* ATCC 29213, multidrug-resistant (MDR) *E. faecalis* VRE10, and MDR *Enterococcus faecium* BM4105 as representative Gram-positive bacteria, and *Acinetobacter baumannii* ATCC 19606, *Escherichia coli* ATCC 25922, *Pseudomonas aeruginosa* ATCC 27853, *Klebsiella pneumoniae* ATCC 13883, MDR *P. aeruginosa* PAO1, MDR *A. baumannii* BAA-1605, and MDR *K. pneumoniae* ATCC 700603 as representative Gram-negative bacteria. The inhibitory effects of the cAMPs were quantified by dividing the cell density ($OD_{600}$) obtained in the presence of AMPs by that obtained in their absence, and the resulting value was expressed as a percentage of growth inhibition. We observed that 84 of the 100 cAMPs inhibited at least one strain ( < 80% growth as used in ProT-Diff[65]), representing a high positive rate of 84% (Fig. 5a, Supplementary Data 17). Simultaneously, a negative set of twenty peptides was chosen from a predicted non-AMP set, of which three were effective in inhibiting at least one strain, indicating that our method has a low false negative rate of 15% (Supplementary Fig. 10a, Supplementary Data 17). Together, these assessments provided meaningful experimental support for model robustness despite the limited scale of peptide synthesis. Notably, most cAMPs had low homology ( < 60%) with known AMPs (Fig. 5b, Supplementary Data 16), indicating that pLLM-based classifiers did not particularly rely on sequence similarity. The 20 cAMPs with the highest activity (appearing within the top 20 cAMPs against at least four strains) were selected for further analysis. These 20 cAMPs were mainly from the kingdom Bacteria (18/20), with Actinomycetota (6/20) and Acidobacteriota (4/20) being the most represented phyla (Supplementary Data 16).

The antibacterial efficacy of the top 20 cAMPs screened from the initial antibacterial assays was further tested through measurements of their minimum inhibitory concentrations (MICs) against commonly studied pathogens, including the Gram-negative bacteria *A. baumannii* ATCC 19606, *E. coli* ATCC 25922, *P. aeruginosa* ATCC 27853, MDR *A. baumannii* BAA-1605, and MDR *K. pneumoniae* ATCC 700603, as well as the Gram-positive bacteria *E. faecalis* ATCC 29212, *S. aureus* ATCC 29213, and MDR *E. faecium* BM4105. Seven cAMPs inhibited the growth of at least one strain at a MIC below 256 $\mu$M. Remarkably, cAMP_81 presented the lowest MICs of 4 $\mu$M, 16 $\mu$M, and 16 $\mu$M against *A. baumannii* ATCC 19606, *E. coli* ATCC 25922, and MDR *A. baumannii* BAA-1605, respectively (Fig. 5c, Supplementary Data 18). However, no cAMPs affected *P. aeruginosa* ATCC 27853 and *E. faecalis* ATCC 29212 at a concentration below 256 $\mu$M. In addition, we synthesized two broad-spectrum AMPs (synechocucin-1[62] and prevotellin-2[61]) as positive controls. We observed that synechocucin-1 and prevotellin-2 displayed antimicrobial activity against seven and three strains, respectively. Notably, compared to positive controls, cAMP_81 displayed lower or comparable MICs against *A. baumannii* ATCC 19606, *E. coli* ATCC 25922, and MDR *A. baumannii* BAA-1605 (Supplementary Fig. 10b).

Given the potential therapeutic application of cAMPs, assessing their cytotoxicity to human cells is critical to ensure safety and selectivity. We conducted hemolysis and cytotoxicity assays on the top 20 cAMPs. The top 20 cAMPs presented low toxicity, with hemolysis rates below 4% in red blood cells, and growth inhibition rates below 10% in L-02 cells and below 30% in 293T cells (Fig. 5e, Supplementary Data 19).

Additionally, we randomly selected and tested 30 other cAMPs, further validating their low toxicity, with hemolysis rates below 5% in red blood cells, and growth inhibition rates below 20% in L-02 cells and below 50% in 293T cells (Supplementary Fig. 11, Supplementary Data 19). These findings demonstrated the high performance of our models in predicting non-toxic peptides. To quantitatively assess the cytotoxicity of cAMPs, we further determined the CC50 (50% Cytotoxic Concentration) values of the seven cAMPs that inhibited at least one strain at a MIC below 256 $\mu$M (Fig. 5c, Supplementary Data 20). All the cAMPs had CC50 values greater than their respective MICs, except for cAMP_40 (Fig. 5d). Taken together, our pLLM-based models identified non-toxic cAMPs with potent anti-bacterial activity and low toxicity from the EEMC.

The secondary structures of peptides are associated with their antimicrobial and other biological activities. To provide structural context without over-claiming, we first predicted the tertiary structures of the 100 synthesized cAMPs and 20 non-AMPs using AlphaFold3[66], and summarized their secondary-structure propensities using the Define Secondary Structure of Proteins (DSSP) algorithm[67]. Among the 100 cAMPs, 89% showed $\alpha$-helical structures, higher than the proportion (7/20, 35%) in 20 non-AMPs (Supplementary Data 16). The predicted structures of 7 cAMPs with MIC values were shown in the Fig. 6a. We further probed the secondary structure tendencies of 7 cAMPs with MIC values through circular dichroism (CD) in water and trifluoroethanol (TFE) mixtures (3:2, $v/v$). We observed 5 helix-dominated peptides (Fig. 6b). CDNN analysis showed that cAMP_15 had the highest $\alpha$-helical fraction (83.3%), while the remaining helix-containing peptides showed almost 30%, and non-helical peptides showed only 17% (Supplementary Fig. 10c). This structural tendency may underlie the membrane-interacting mechanisms commonly observed in $\alpha$-helical antimicrobial peptides[30,40].

We then focused on investigating the mechanisms of action for selected cAMPs. We first used a transmission electron microscope (TEM) and a scanning electron microscope (SEM) to monitor potential morphological changes of the bacteria upon the exposure to cAMP_81, due to its great antimicrobial efficacy. Four untreated bacteria showed spherical shapes with smooth surface, whereas cAMP_81-treated cells showed marked morphological changes, including membrane disruption and wrinkled surface structures (Fig. 6c–f). Subsequently, three representative peptides (cAMP_81, cAMP_48 and cAMP_102) that exhibited low MICs against *A. baumannii* ATCC 19606 (Fig. 5c) were selected for further analysis using membrane integrity assay. Propidium iodide (PI) could bind to intracellular chromatin, emit fluorescence and indicate membrane damage. We observed the PI fluorescence of *A. baumannii* treated with cAMPs was 2-14 times that of untreated controls (Fig. 6g). This result illustrates that cAMPs may inhibit bacteria by breaking the cytoplasmic membrane, and show dose-dependent effects. Notably, longitudinal resistance assays with cAMP_81 against *A. baumannii* ATCC 19606, *E. coli* ATCC 25922, MDR *E. faecium* BM4105, and MDR *A. baumannii* BAA-1605, using polymyxin B as the positive control, demonstrated that 0.5 × MIC cAMP_81 treatment for 30 days induced less resistance compared with the positive control (Supplementary Fig. 10d, Supplementary Data 21). In summary, these results suggest that the cAMPs identified from the EEMC exert their antimicrobial effects primarily through disruption of the bacterial cytoplasmic membrane, and the low resistance induction observed in the longitudinal assays further supports their potential as membrane-targeting antimicrobial agents.

## Discussion

Extreme environment microbiomes are considered a rich source of novel microbes and functions, yet they remain underexplored at a global scale. Here, to our knowledge, we present the first genome and gene catalog of microbes from diverse extreme habitats, comprising over 78,000 archaeal and bacterial genomes and nearly 4 billion non-

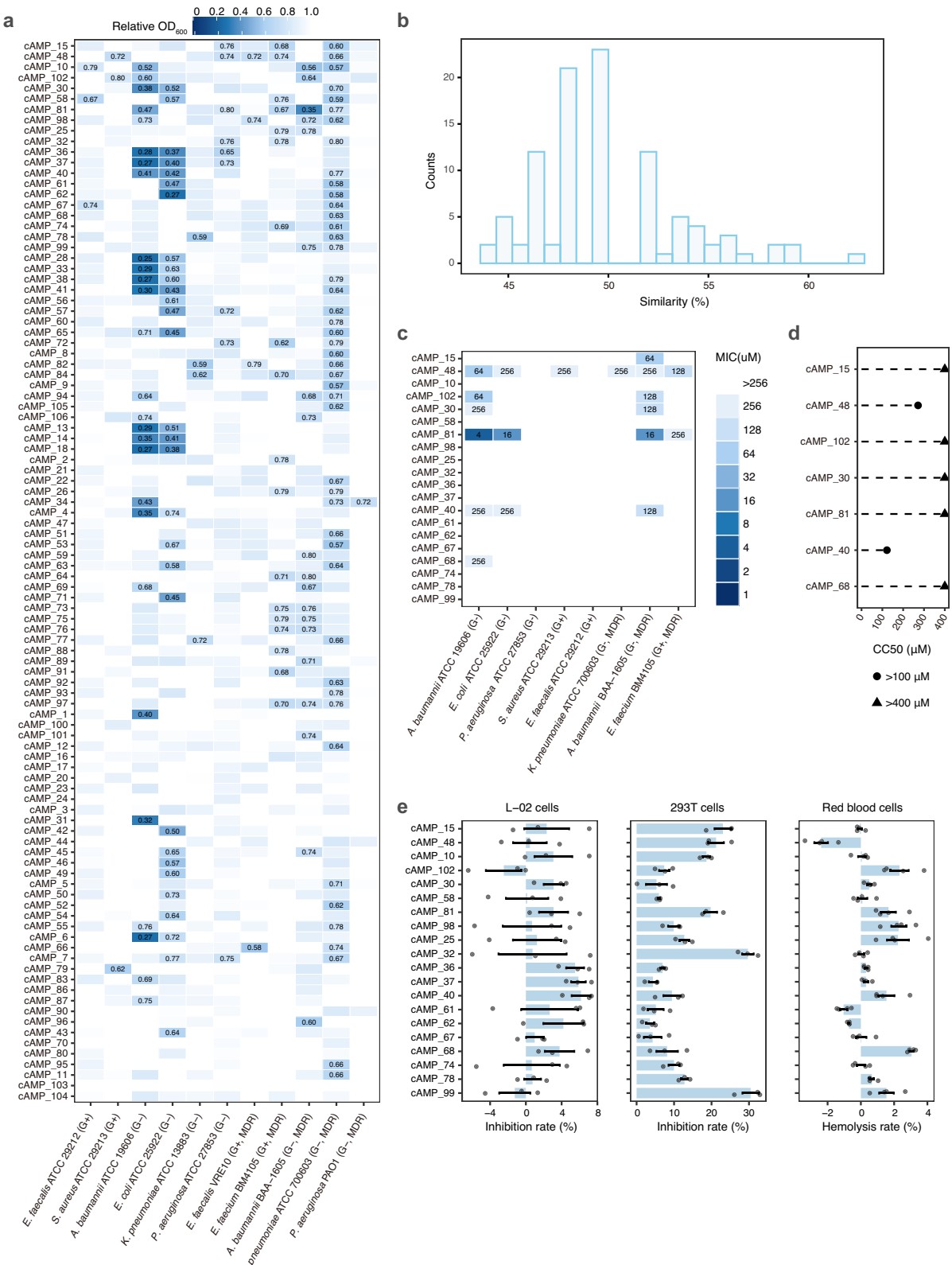

redundant genes. The EEMC substantially expands the known microbial diversity, increasing species and unigene numbers by 18.22% and 19.21%, respectively. The EEMC also shows remarkable biosynthetic potential, with 163,693 BGCs identified, over 50% of which are classified as novel. Furthermore, by integrating deep learning models and experimental validation, we confirm the discovery of diverse, non-toxic AMPs from the EEMC, highlighting its potential for biomedicine

applications. Collectively, the EEMC provides substantial taxonomic and functional novelty, and our study offers a reproducible pipeline to support future drug discovery efforts.

We integrate microbial genomes obtained using cultivation-dependent and cultivation-independent methods, and cluster them into 32,715 representative species, with 63.00% considered novel, highlighting the high diversity and novelty of microbial populations

**Fig. 5 | Experimental validation and potency assays of cAMPs. a** The heatmap shows the relative $OD_{600}$ values of the cells grown in medium supplemented with cAMPs at a concentration of 60 $\mu$M, normalized to the $OD_{600}$ of the cells grown in unsupplemented medium. Only the relative $OD_{600}$ below 80% is labeled.
**b** Distribution of highest similarity between 100 synthesized cAMPs in our study to AMPs of the training dataset. Most of our cAMPs have less than 60% similarity to previously known AMPs in the training dataset. **c** Determination of MICs of the top 20 cAMPs against eight commonly studied pathogens. The heatmap shows the MIC values obtained within the 2-fold dilution concentration range studied. Gram-positive (G+), Gram-negative (G−), and MDR bacteria are indicated (bottom). **d** The CC50 values of the seven selected cAMPs were determined by using the CCK-8 assay on L-02 cells with the cAMPs at a range of concentrations. **e** The results of cytotoxicity tests and hemolysis assays for the top 20 cAMPs at a concentration of 60 $\mu$M. The inhibition rates ($n = 3$ biological replicates) and hemolysis rates ($n = 4$ biological replicates) are presented as the mean values ± s.e.m. Source data are provided in this paper. cAMP candidate non-toxic antimicrobial peptide, MIC minimum inhibitory concentration, MDR multidrug resistant.

within the EEMC. The gene-level novelty further exemplifies this diversity and adaptation to extreme environments, as the rarefaction curve did not reach saturation at the current sequencing depth. Moreover, the limited species-level overlap (0.73%) between MAGs and isolate genomes underscores their complementarity. Notably, we generated 83 isolate genomes from deep-sea samples, yielding 42 representative and 22 novel species, preserving genetic resources digitally and facilitating future investigations.

We further characterize the diversity and distribution of BGCs obtained from the EEMC. We observe that the biosynthetic potential varies across different environments and taxonomic groups, suggesting that the microbes may produce structurally and functionally diverse secondary metabolites to adapt to extreme conditions. RiPPs accounted for the largest proportion of BGCs in the EEMC ($n = 46,816$, 28.60%), underscoring their potential as a major source of AMPs. Notably, members of well-studied producers (such as Pseudomonadota, Actinomycetota, *Streptomyces*, *Mycobacterium* and *Micromonospora*[19,24,57,68]) and unexplored taxonomic groups (such as *JAKAGC01*) encode exceptionally diverse BGC repertoires. Rarefaction analysis indicates that the current sampling has not yet saturated the biosynthetic space, pointing to substantial undiscovered potential as more microbes from extreme environments are sequenced.

Despite the scale and novelty of the EEMC, certain limitations remain. First, the known isolate genomes from extreme environments remain limited (only around 3,000), highlighting the need for more extensive isolation efforts. Such cultivation can be guided by metabolic traits inferred from MAGs to better capture microbial diversity[6,69]. Second, the majority of BGCs are fragmented, especially those from MAGs (88.45% versus 30.18% in isolate genomes), due to the limitations of short-read sequencing and assembly[70,71]. This fragmentation, coupled with the dependence of rule-based tools like antiSMASH on known domain architectures and highly contiguous genomes[72,73], limits the completeness and accuracy of BGC prediction and likely leads to an underestimation of biosynthetic diversity, thereby obscuring novel secondary metabolites. Third, although the gene catalog captures substantial sequence novelty, the enzymes with potential for biotechnological applications, such as DNA polymerase and xylanases, have not yet been fully explored[2,74]. Future efforts integrating long-read sequencing, deep learning methods for functional annotation and BGC prediction, targeted isolation and cultivation guided by MAG-derived metabolic insights, and experimental validation of predicted functions will be essential to fully realize the potential of the EEMC. Nonetheless, the scale and novelty of the EEMC ensure its value as a foundational global reference for microbial diversity and biosynthetic capacity.

Given this extensive biosynthetic repertoire, we next focused on exploring antimicrobial peptides from EEMC using deep learning-based approaches. We constructed three pLLM-based classifiers and observed that each consistently outperformed conventional[32,32,34,37] and prior microbiome-mining AMP predictors[30]. This underscores the increasing utility of large-scale protein language models in decoding complex sequence-function relationships, as has been previously demonstrated in structure prediction and enzyme function annotation[43,75,76]. To address the challenge of peptide toxicity in AMP

development[30,40], we incorporated toxicity prediction modules into our framework. Although our toxicity classifiers did not surpass ToxinPred in benchmark accuracy, they correctly identified two non-toxic peptides (cAMP_17 and cAMP_51) misclassified by ToxinPred. This suggests that conventional models may struggle with novel or extremophile-derived sequences, where data are sparse and sequence features deviate from canonical non-toxic peptides. In contrast, our pLLM-based models, leveraging broader semantic representations, better captured these outliers. These findings underscore the value of deep learning approaches tailored to novel sequence space in AMP discovery from extreme environment microbiomes.

Follow-up experimental validation confirmed the ability of our classifiers to discover diverse non-toxic AMPs from the EEMC. We predicted 3032 non-toxic cAMPs from 11,379 core peptides, and synthesized 100 candidates, 48 of which originated from previously uncharacterized taxonomic lineages. Among these 100 cAMPs, 84 showed inhibitory activity against at least one pathogen, and most shared low sequence similarity ( < 60%) with known AMPs (Supplementary Data 16), suggesting that the predictions were not driven by simple sequence homology. Notably, these peptides were generally more effective against Gram-negative bacteria and showed strain-specific activity, consistent with the identified AMPs in human microbiomes[30,61]. We further observed that seven cAMPs with measurable MICs displayed distinct capacities to disrupt cytoplasmic membrane, five of which exhibited $\alpha$-helical structure, suggesting potentially diverse antibacterial mechanisms. In addition, we identified 6 cAMPs derived from archaeal BGCs, one of which (cAMP_102) exhibited MICs of 64 $\mu$M and 128 $\mu$M against *A. baumannii* ATCC 19606 and MDR *A. baumannii* BAA-1605, respectively. Recent work has revealed archaeal proteomes as a rich source of bioactive peptides[39], underscoring the untapped potential of the archaeal microbiomes in antibiotic discovery. Together, these findings expand the known phylogenetic and ecological spectrum of AMP-producing lineages and open avenues for future studies to mechanistically characterize these peptides and explore their structure-activity relationships.

While the MAI framework represents a major advancement over previous approaches, it can be further extended in several ways. First, the MAI framework relies on sequence embeddings without explicit incorporation of structural or physicochemical modeling. The pLLMs we employed, including ESM series and ProteinTrans, were black-box models by lacking interpretability, which hinders feature characterization of bioactive peptides. Integrating three-dimensional structure-informed representations derived from popular models, such as AlphaFold3[66], may further improve the performance of the MAI framework and model interpretability. Additionally, to achieve more high-throughput filtering of peptides and improve reliability, the stability of predicted structures and peptide-membrane interactions remain to be validated based on physically based methods, such as molecular dynamics simulations[35,36]. The MAI framework is also well-suited for the development and iterative optimization of AMPs with more specific characteristics, such as antimicrobial selectivity and specificity, serum stability, and in vivo bioactivity.

We acknowledge that the peptides synthesized in this study may differ significantly from their native counterparts in folding, stability, and bioactivity. The pLLM-based model was trained exclusively on

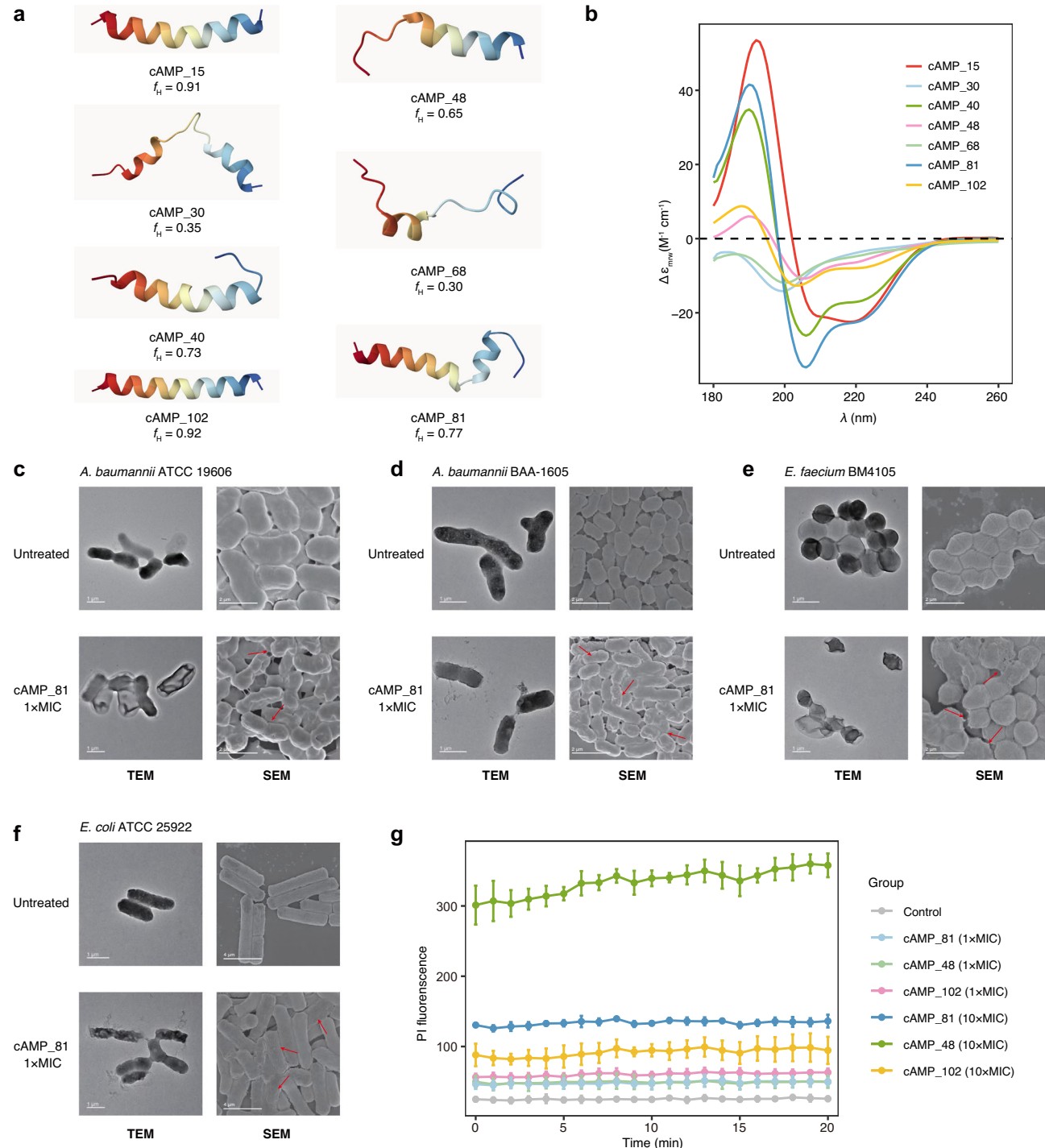

**Fig. 6 | Determination of peptide structures using CD spectra and mechanism of action studies of cAMPs. a** AlphaFold3 was used to generate structural predictions using default parameters. Three-dimensional structures of the resulting PDB files were generated using Mol*3D Viewer. **b** CD results for the seven cAMPs with MIC values. **c–f** TEM and SEM examination of four bacterial strains treated with cAMP_81 at 1 × MIC concentration and non-AMP negative control group, revealing leakage of cell contents and disruption of the cell wall and membrane. Experiments were performed in triplicate with similar results and one representative figure is shown. **g** PI assays against *A. baumannii* ATCC 19606 for three selected cAMPs were used to examine the potential mechanism of function of cAMPs at 1 × MIC and 10 × MIC concentration. Data are presented as mean values ± SD (*n* = 3 biological replicates). Source data are provided in this paper. CD circular dichroism, TEM transmission electron microscope, SEM scanning electron microscope, MIC minimum inhibitory concentration, PI propidium iodide.

amino acid sequences without structural or modification annotations, and the core peptides predicted by DeepRiPP represent the genetically encoded regions prior to enzymatic processing. Consequently, the outputs of the MAI framework represent unmodified peptide scaffolds rather than mature RiPPs. This decision reflects both the current uncertainty in accurately inferring PTM patterns and sites from precursor sequences alone[27,29,64] and the prohibitive cost of chemoenzymatic or total chemical installation of PTMs at the scale of this study (96 of the 100 synthesized cAMPs were lassopeptide or class II lanthipeptide core peptides).

Although PTMs and stereochemical constraints are often essential to the native bioactivity of RiPPs, the goal of the present study was not to reconstruct the full biosynthetic maturation of RiPP natural products but rather to evaluate the antimicrobial potential encoded in peptide backbones derived from BGCs. Importantly, RiPP-type BGCs provide a biologically meaningful starting point for peptide discovery, as they encode genetically templated precursor peptides that occur within defined biosynthetic contexts and are associated with dedicated modification enzymes. Even in the absence of PTMs, the core peptide sequences represent a structured and biologically derived sequence space that serves as a high-quality, non-random library for peptide discovery, which may have contributed to the high success rate (84/100) in identifying active peptides. Moreover, the resulting unmodified scaffolds are synthetically accessible via standard peptide synthesis, making them readily amenable to further optimization and functional characterization. Screening peptides derived from such genomic contexts therefore provides a rational strategy to explore naturally encoded antimicrobial scaffolds. Nevertheless, the absence of PTMs likely restricts the accessible chemical space of the synthesized peptides and may partly explain the observed convergence in modes of action of cAMPs. Going forward, as curated PTM datasets become more standardized and accessible, incorporating such information with deep learning methods would enable the prediction of enzymatic PTM sites, validation of stereochemistry, and comparisons between modified RiPPs and their unmodified cores.

In conclusion, the EEMC provides an unprecedented genomic and biosynthetic resource that captures the global taxonomic, functional, and adaptive diversity of extremophilic microbiomes. The EEMC significantly extends the known microbial and natural product landscape, serving as a valuable foundation for exploring life at the limits of Earth's biosphere. Beyond deepening our understanding of microbial adaptation and secondary metabolism, the resource could facilitate the discovery of extremozymes, novel antibiotics, and stress-resilient functions with translational potential in biomedicine, industry, and environmental sustainability.

## Methods

### Metagenome sequence data collection, assembly, and binning

We collected and reanalyzed metagenomic sequencing data from samples collected from extreme environments with available metadata published from January 2010 to February 2024 (Supplementary Data 1). We first screened published studies in the NCBI PubMed database, by searching keywords related to various extreme environments and ecosystems. We cross-referenced the identified datasets and removed duplicates. We further checked and removed samples of rRNA gene amplicon sequences, metatranscriptomic sequences and host-associated sequences. A total of 2293 metagenomic samples with more than 23 Tb sequencing data were downloaded from the NCBI and NGDC databases.

Raw metagenomic sequencing reads were obtained using sra-toolkit (version 3.0.0) and transformed to fastq format using fastq-dump (version 2.9.6). Reads with low sequencing quality and adapter sequences were removed using fastp with default parameters (version 0.12.0)[77]. Clean data of each metagenomic sample were assembled by MEGAHIT (version 1.2.9) with the parameters "–presets meta-sensitive"[78]. The assembled contigs were binned and refined using MetaWRAP (version 1.3.2) with default parameters[79], which integrated the binning results of metaBAT2 (version 2.12.1)[80], MaxBin2 (version 2.2.7)[81], and CONCOCT (version 1.1.0)[82]. A total of 77,146 refined MAGs were further refined using MAGpurify (version 2.1.2) with default modules and the parameters "phylo-markers", "clade-markers", "tetra-freq", "gc-content", and "known-contam" to remove contaminated contigs originating from a different species than the dominant organism in the MAG[83]. This purification process resulted in the removal of a total of 236,681 contigs, accounting for 0.74% of the initial 32,034,758 contigs within the MAGs.

### Sampling, sequencing, assembly and collection of isolates

To broaden the genome catalog, isolate genomes were included in downstream analyses. A total of 3131 publicly available bacterial and archaeal isolate genomes derived from extreme environments were downloaded from the NCBI RefSeq genome database[44]. Specifically, we downloaded the available metadata of 350,267 genomes from the database using the NCBI Datasets (up to August, 2024), which is useful to download biological sequence data across all domains of life from NCBI. The high quality genomes (completeness ≥95% and contamination <5%) were selected if their "isolation_source" or "geo_loc_name" matched the regular expression of any extreme environment (Supplementary Data 22). We further curated the dataset by manually removing genomes derived from humans or other species.

In addition, we derived bacteria from cold seep sediment and generated in-house isolate genomes. Briefly, sediment samples were retrieved from Haima cold seep field, northern South China Sea in August 2023. The sediment samples were resuspended in sterile seawater and mixed thoroughly. The samples were subsequently diluted using gradient dilution method, and the dilutions were spread on 2216E medium (Park Hope, Shandong, China) and Gauze's Synthetic Medium No.1 (Park Hope, Shandong, China). The plates were then incubated in an inverted position at 28 °C. After several days of culture, single colonies were selected and transferred to new 2216E plates for purification. The isolates were then inoculated into glycerol tubes for storage at −80 °C. The cultivated isolates were cultured in 2216E broth at 28 °C, and cells were harvested by centrifugation at 4000 × g for 10 minutes at the mid-log phase. Genomic DNA was extracted from the cultures using a TIANamp Bacteria DNA Kit (Tiangen, Beijing, China) and then sequenced by BGI using the DNBSEQ-T7 platform with 150-bp paired-end reads. Fastp (version 0.12.0) was used for quality control of the paired-end reads; specifically, adapter sequences and low-quality reads were removed. The genomes of the cultivated isolates were assembled using Unicycler (version 0.5.0) with default parameters, and only contigs ≥500 bp were retained[84]. Unicycler functions as a SPAdes-optimiser when applied to short-read only sets.

### Quality evaluation of MAGs and isolate genomes

The completeness and contamination of each purified MAG and isolate genomes were evaluated using the module "lineage_wf" in CheckM (version 1.2.1)[85]. The presence of ribosomal RNAs (rRNAs) and transfer RNAs (tRNAs) was identified using Infernal (version 1.1.15) with the parameters "–cut_ga, –rfam"[86] and models from the Rfam database (version 14.10)[87]. For MAGs, we retained only assemblies meeting the MIMAG medium- and high- quality standards[45], which were defined as follows: 1) medium-quality MAGs: completeness ≥50% and contamination <10%; and 2) high-quality MAGs: completeness >90% and contamination <5% with the presence of the 23S, 16S and 5S rRNA genes and at least 18 tRNAs. For isolate genomes, assemblies with completeness ≥95% and contamination <5% were retained. Retained genomes were further assigned a QS, calculated as completeness - 5 × contamination, following the implementation in the Tibetan Plateau Microbial Catalog (TPMC)[16].

### Clustering MAGs and isolate genomes into species-level OTUs

The 78,213 genomes were clustered into 32,715 representative species-level OTUs using the module "dereplicate" in dRep (version 3.4.3)[88], on the basis of an AF threshold of 30% and a genome-wide ANI threshold of 95% with the parameters "-nc 0.3, -sa 0.95, -comp 0, -con 1000" as used in the TPMC[16].

## Comparing representative OTUs to the genomes within public catalogs

The 32,715 representative OTUs were compared to 22,732 representative MAGs of environmental origin from GEM[47], 24,195 representative genomes from GOMC[19], and 957 representative MAGs from TARA Ocean[46], using the module "compare" in dRep. The OTUs exhibiting an ANI < 95% with the compared genomes were designated as novel.

## Taxonomic annotation and phylogenetic tree construction

Taxonomic annotation of 32,715 representative OTUs was performed using the module "classify_wf" in the Genome Database Taxonomy Toolkit (GTDB-Tk, version 2.4.0) with default parameters against the GTDB release R220[89]. The GTDB release R220 comprises 113,104 representative genomes, 107,235 of which are bacterial species and 5869 of which are archaeal species. The phylogenetic trees based on 122 archaeal or 120 bacterial marker gene sets were generated using GTDB-Tk and visualized using the R package "ggtree".

## ORF prediction and non-redundant gene catalog construction

Gene ORFs were predicted from metagenomic contigs using Prodigal (version 2.6.3)[90] with the parameters "-p meta". All ORFs were clustered into non-redundant gene clusters at a threshold of 95% identity and 80% coverage, using MMseqs2 (version 113e321)[91] with the parameters "easy-linclust -e 0.001 –min-seq-id 0.95 -c 0.80", as used in the TPMC[16]. The gene rarefaction analysis for each extreme environment was performed based on the basis of the MMseqs2 gene clustering results using an in-house Python script, with random sampling at 10% intervals 100 times.

## Gene catalog annotation

For functional annotation, the non-redundant genes were aligned against the NR databases (https://ftp.ncbi.nlm.nih.gov/blast/db) using DIAMOND (version 2.0.15)[92] with the parameters "blastp e-value 1e-5", and against the UniRef50 and Swiss-Prot databases[48,49] using MMseqs2 with the parameters "easy-search -e 0.01 –min-seq-id 0.3 –cov-mode 2 -c 0.8 –max-seqs 1". The MMseqs2 parameters follow those used in the TPMC dataset[16], except for setting max-seqs to 1 to save storage resources. Additionally, the non-redundant genes were aligned against the eggNOG Orthologous Groups database (version 5.0)[93] for functional annotation, using eggNOG-mapper (version 2.1.10)[94]. The eggNOG database integrates functional annotations collected from several sources, including the Cluster of Orthologous Groups (COG) categories database[50], the KEGG functional orthologs[51], the carbohydrate-active enzymes (CAZy) database[52], and the Gene Ontology (GO) resource[53]. Furthermore, ARGs and VFs were annotated by aligning the protein sequences of the non-redundant genes against the Antibiotic Resistance Database (CARD)[54] and VFDB (version 2024)[55], respectively, using DIAMOND (version 2.0.15) with the parameters "blastp e-value 1e-5". The genes that did not align with any of the above databases were considered to be novel genes.

## Comparison of genome-derived genes with those from public datasets

We used Prodigal with default parameters for ORF prediction on the MAGs and isolate genomes from the extreme environments, as well as the genomes from the GEM and GOMC datasets[19,47]. The protein sequences of ORFs from the above three datasets were combined and clustered into gene clusters at a threshold of 70% identity and 80% coverage, using MMseqs2 with the parameters "easy-linclust -e 0.001 –min-seq-id 0.7 -c 0.8", as used in the TG2G[18]. The VFs, ARGs, and CAZymes were annotated by aligning the protein sequences of all above ORFs against the VFDB, CARD, and CAZy database, respectively, using DIAMOND with the parameters "blastp e-value 1e-

5 id 40 query-cover 80"[95]. MMseqs2 was used to evaluate the similarity of the VFs from extreme environments with those in the GEM and GOMC datasets. The same approach was applied for ARGs and CAZymes.

## BGC identification and clustering

A total of 163,693 BGCs were predicted and identified on contigs ≥5 kb of the 78,213 genomes to reduce the risk of fragmentation, using antiSMASH (version 7.0.1) with the parameters "–minlength 5000"[72]. These BGCs were categorized into eight groups using BiG-SCAPE (version 1.1.5)[96]: "PKSI", "PKSother", "NRPS", "RiPPs", "Saccharides", "Terpene", "PKS-NRP_Hybrids", and "Others". The BGCs were further clustered into 64,733 gene cluster families (GCFs, 0.3 distance threshold) and 2178 gene cluster clans (GCCs, 0.7 distance threshold) using BiG-SCAPE with default parameters. To avoid sampling biases in quantitative analyses (taxonomic and functional compositions of GCCs/GCFs, as well as GCF and GCC distances to reference databases), the 163,693 BGCs were further dereplicated by retaining only the longest BGC per GCF per species, leading to a total of 107,829 BGCs.

## Novelty of BGCs, GCFs, and GCCs

The novelty of the BGCs, GCFs, and GCCs was estimated by querying the 107,829 representative BGCs against a preprocessed BiG-FAM reference database, using BiG-SLiCE (version 1.1.0) with default parameters[97]. This BGC reference database contains 1,225,071 BGCs from 209,206 publicly available microbial genomes and metagenome-assembled genomes, which integrates MIBiG v2.0, RefSeq complete/draft bacteria, GenBank fungi, GenBank archaea, and other MAGs from different studies; 29,955 GCF reference models have been generated from the reference database. The distance value was calculated for each representative BGC and the assigned GCF model to indicate how closely they matched[97]. The query BGCs with distances ≤ 900 as used in BiG-FAM were defined as matched BGCs, whereas BGCs with distances > 900 were defined as novel BGCs. The GCFs with a proportion of matched BGCs < 0.2 were considered novel GCFs, whereas the GCCs with a proportion of matched BGCs < 0.4 were considered novel GCCs.

## Peptides collection and preprocessing

AMP data were collected from seven public AMP databases, including APD[98], CAMP[99], DBAASP[100], DRAMP[101], dbAMP[102], LAMP2[103], and BaAMP[104] (downloaded on August, 2024), and merged into a single dataset. These databases included both synthetic and naturally derived peptides, and the antimicrobial activity data included both in vitro assays and in vivo experiments. To ensure adequate sample size, we did not stratify peptides by origin or experimental context.

Sequences that included nonessential amino acids or were longer than 300 amino acids were removed, resulting in a dataset of 62,182 AMP sequences. We then removed the duplicate sequences from the dataset, resulting in 28,449 non-redundant AMPs (Supplementary Data 13), which include 16,061 anti-Gram+ peptides, 16,451 anti-Gram- peptides, 7068 antifungal peptides, 2770 antiviral peptides, and 6771 peptides labeled as antibacterial or antimicrobial. The non-AMP dataset was downloaded from UniProt (https://www.uniprot.org) by setting the "Sequence length" from 0 to 300 and removing any entry that matched one of the following keywords: antimicrobial, antibiotic, antiviral, antifungal, effector, excreted, toxic, anticancer, defensing, AMP, membrane, and secretory (downloaded on August, 2024). We then removed the duplicate sequences in the dataset and peptides with non-essential amino acids. Additionally, sequences identical to any AMPs were removed from the non-AMP dataset, yielding a total of 166,448 non-AMP sequences.

To accommodate model training while considering the feasibility of downstream synthesis, we retained only peptide sequences with a

length of 100 amino acids or less. This threshold balances the need for sufficient sequence context during representation learning with the practical limitations of peptide synthesis (usually ≤ 30 amino acids). The structure of the remaining datasets is as follows:

- Global AMP: A total of 55,209 sequences (25,004 AMPs, 30,205 non-AMPs)
- Anti-G+ AMP dataset: A total of 45,069 sequences (14,864 AMPs, 30,205 non-AMPs)
- Anti-G- AMP dataset: A total of 45,493 sequences (15,288 AMPs, 30,205 non-AMPs)

Toxic and non-toxic peptides were collected from three public databases: Hemolytik[105], DBAASP[100], and ToxinPred3.0[63]. Most peptides lacked explicit metadata regarding their biological origin or experimental context; therefore, to maintain sufficient data diversity, we did not stratify peptides by source or experimental context. Duplicate sequences were removed from the toxic peptide dataset, yielding 6518 non-redundant toxic peptides. Similarly, duplicate sequences were eliminated from the non-toxic peptide dataset, resulting in 6843 non-redundant non-toxic peptides. Furthermore, sequences shared between the toxic and non-toxic datasets were excluded, resulting in a final dataset of 6318 toxic peptides and 6643 non-toxic peptides.

The distributions of peptide properties, including sequence length, isoelectric point, charge, hydrophobic ratio, aromaticity, and amino acid composition, are provided (Supplementary Figs. 12 and 13).

## Model training
Instead of words representing tokens that form sentences in the human language, in our use case, we deal with amino acids forming protein sequences. Each collected dataset mentioned previously was randomly split into 10 equal parts: eight parts were selected as the training set, one part was used for validation, and the remaining part was used for testing. Training of the pLLMs is typically performed using a masked language modeling task, where the input is generated by randomly corrupting 5–15% of the original amino acids in a protein sequence, and the model is trained to reconstruct the original sequence. Existing pLLMs trained in this manner include ProtT5-XL-UniRef50 (PTrans)[41], ESM2[42], and ESM3[43]. We adopt a probing strategy[106] in which the encoder of a pretrained pLLM is leveraged to extract fixed-length embeddings from the last hidden layer for each protein sequence. These embeddings are then fed into a downstream classifier head to infer properties such as antimicrobial activity or toxicity. Once trained, the model can predict these properties for novel protein sequences derived from extreme environments or any other sources.

To maximize the utility of existing pLLMs, we developed the MAI framework. This framework incorporates the pretrained pLLM backbone (referred to as the protein language model (PLM) component) and a user-configurable classification head composed of linear and ReLU layers. When the PLM component produces sufficiently discriminative embeddings, a simple classifier is often adequate for downstream tasks. To improve efficiency, MAI supports freezing the PLM component during training and caching the embeddings it generates. This enables the reuse of precomputed embeddings in subsequent training steps, significantly reducing computational time. The entire trained model can be saved and deployed for future prediction tasks.

## Prediction of RiPP core peptides and cAMPs
The coding sequences of 46,816 RiPP-type BGCs were extracted from the GenBank files generated by antiSMASH, and subsequently analyzed using DeepRiPP[29] to predict putative core peptides. The putative core peptides were further evaluated for antimicrobial activity and toxicity using MAI framework.

## Sequence similarity estimation
We applied a global pairwise sequence alignment using the Needleman-Wunsch algorithm with the Bio.pairwise2 module in Python (version 3.9) to estimate the similarity between 11,379 RiPP core peptides and peptides in two training datasets. The sequence identity for each alignment was normalized by the maximum length of the query or subject sequence. For each query, the subject sequence with the highest identity was recorded as the best hit.

## Peptide synthesis
All the cAMPs, non-AMPs, and two published AMPs (synechocucin-1 and prevotellin-2) used in this study were synthesized via solid-phase peptide synthesis by Sangon Biotech. The molecular masses were determined by mass spectrometry. The purity of each peptide was determined using high-performance liquid chromatography, with all peptides exhibiting a purity greater than 95%.

## Bacterial inhibition experiment
Eleven bacterial strains, including *E. faecalis* ATCC 29212, *S. aureus* ATCC 29213, MDR *E. faecalis* VRE10, MDR *E. faecium* BM4105 as representative Gram-positive bacteria, and *A. baumannii* ATCC 19606, *E. coli* ATCC 25922, *P. aeruginosa* ATCC 27853, *K. pneumoniae* ATCC 13883, MDR *P. aeruginosa* PAO1, MDR *A. baumannii* BAA-1605, and MDR *K. pneumoniae* ATCC 700603 as representative Gram-negative bacteria, were selected for bacterial inhibition assays. *A. baumannii* ATCC 19606, *A. baumannii* ATCC BAA-1605, *K. pneumoniae* ATCC 13883, and *K. pneumoniae* ATCC 700603 were purchased from Beinuo Biotechnology (Shanghai, China), originally sourced from the ATCC. Additional bacterial strains used in this study were kindly provided by collaborating laboratories as listed in the Acknowledgments. The strains were streaked on Luria-Bertani (LB) (Thermo Fisher Scientific, Waltham, USA) agar plates and incubated at 37 ℃ overnight. The single colonies were transferred to LB media and cultured at 37 ℃ with shaking at 120 × g overnight. The LB bacterial suspension was adjusted to a starting concentration with an $OD_{600}$ of 0.1, and then diluted 1000 times for the inhibition assays. Freeze-dried powders of cAMPs and non-AMPs were first dissolved in double-distilled water to a final concentration of 2.4 mM. We set three groups to test cAMP antibacterial activity: (1) blank group, with 200 $\mu$l of LB medium; (2) negative control group, with 100 $\mu$l of LB medium, 95 $\mu$l of bacterial suspension, and 5 $\mu$l of sterile water; and (3) test group, with 100 $\mu$l of LB medium, 95 $\mu$l of bacterial suspension, and 5 $\mu$l of cAMP mother solution, with a final concentration of cAMP of 60 $\mu$M. Experiments were performed in 96-well plates with a working volume of 200 $\mu$l. The $OD_{600}$ value of each well was measured after 12 h of cultivation at 37 ℃. Three technical replicates were performed for all experiments. The % growth of each cAMP group was calculated as $(OD_{600\,test} - OD_{600\,blank})/(OD_{600\,negative\,control} - OD_{600\,blank})$.

## MIC determination
The MIC of each peptide was determined using the broth microdilution method[107]. Briefly, *E. faecalis* ATCC 29212, *S. aureus* ATCC 29213, *E. coli* ATCC 25922, *P. aeruginosa* ATCC 27853, *A. baumannii* ATCC 19606, *A. baumannii* BAA-1605, *E. faecium* BM4105, and *K. pneumoniae* ATCC 700603 were inoculated in Mueller-Hinton Broth (MHB) (Thermo Fisher Scientific, Waltham, USA) at 37 ℃ overnight. The cultures were diluted with fresh MHB and adjusted to 5 × 10⁵ CFU/ml. The peptides diluted in water were added to 96-well plates at 2-fold dilutions ranging from 256 to 1 $\mu$M. Then, 100 $\mu$l bacterial cultures were evenly added to the plates, and incubated at 37 ℃ overnight. The MIC was determined as the the minimum concentration of peptides at which no bacterial growth was visible.

## Hemolysis assay
Sheep blood cells (Defibrinated Sheep Blood, BkmamLab, Changde) were centrifuged at 1,500 × g to obtain red blood cells, which were then

washed and resuspended in PBS to 4% ($v/v$). A suspension of red blood cells was added to 96-well plates. Blanks and 1% Triton-X100 (Sigma-Aldrich, Germany) containing PBS were used as negative and positive controls, respectively. The cAMPs were added to the wells at a final concentration of 60 $\mu$M. The samples were incubated for 1 h with gentle shaking and then were centrifuged at 1500 × g for 10 min at 4 °C. The supernatant from each well was transferred to new 96-well plates. The release of hemoglobin was detected at 570 nm with a Multiskan MK3 microplate reader (Thermo Fisher Scientific, Waltham, USA). The hemolysis rate for each peptide was calculated as ($OD_{570\ peptides}$ − $OD_{570\ negative\ control}$)/($OD_{570\ positive\ control}$ − $OD_{570\ negative\ control}$).

### Cytotoxicity assays against mammalian cells
CCK-8 assays were performed to assess the cytotoxicity of the cAMPs in vitro. L-02 (BNCC, Beijing, China) and 293T (Procell, Hubei, China) cells were seeded into each well of 96-well plates. After 24 h incubation at 5% $CO_2$ and 37 °C, the medium was replaced with fresh medium, and cAMPs at various concentrations were added to the wells, followed by 48 h of incubation. Afterward, the cells were washed, 10% CCK-8 solution (Biosharp, Anhui, China) was added, and the mixture was incubated for 3 h. The absorbance of the samples was measured at 450 nm with a Multiskan MK3 microplate reader (Thermo Fisher Scientific, Waltham, USA). Cell viability was calculated as ($OD_{450\ control}$ − $OD_{450\ peptides}$)/($OD_{450\ control}$ − $OD_{450\ blank}$). The CC50 value of each cAMP was calculated using GraphPad Prism software (version 8.0.2; GraphPad Software, Inc.).

### Circular dichroism assays
CD experiments were performed using a Chirascan V100 (Applied Photophysics, England) at the Research Center for Deepsea Bioresources, Sanya. The experiments were carried out at a temperature of 25 °C in triplicates. Spectra collections were obtained using a quartz cuvette with an optical path length of 0.5 mm, covering a wavelength range from 260 to 190 nm at a rate of 50 nm/min and a bandwidth of 1 nm. The peptides were tested at a concentration of 125 $\mu$M. Measurements were performed in a mixture of water and TFE at a ratio of 3:2 ($v/v$). Baseline measurements were recorded prior to each measurement. To minimize background effects, a Fourier transform filter was applied. The helical fraction and other fraction values were calculated using the CDNN (version 2.1)[67].

### TEM and SEM
Bacterial cell envelope damage induced by cAMP_81 was visualized by TEM and SEM. Four pathogens (*A. baumannii* ATCC 19606, *E. coli* ATCC 25922, MDR *A. baumannii* BAA-1605, and MDR *E. faecium* BM4105) were cultured to the exponential phase ($OD_{600}$ = 0.5) at 37 °C. Then, the bacterial cells were harvested by centrifugation at 4,000 × g for 5 min at room temperature and resuspended in 5 mM HEPES buffer (pH 7.0) containing 5 mM glucose to an $OD_{600}$ of 1.0. The bacterial suspensions were treated with cAMP_81 at a final concentration of 1 × MIC at 37 °C for 5 h. Untreated bacterial suspension was used as the control. For TEM, the 300-mesh carbon-coated Cu grids were glow discharged for 30 s immediately prior to use. Grids were incubated on 10 $\mu$l of bacterial suspensions for 1 min before blotting the media off with filter paper. Each grid was then dipped in 5 $\mu$l of distilled water and blotted in between transfer to new droplets. Each grid was subsequently incubated in 5 $\mu$l of 0.05% uranyl acetate for 30 s before blotting and left to dry before image acquisition. The samples were observed using a JEM-F200 electron microscopy (JEOL, Japan). For SEM, bacteria were fixed at 4 °C with PBS containing 2.5% glutaraldehyde (Sigma-Aldrich, Germany) for 4 h. Then, the fixed samples were dehydrated using a series of ethanol solutions (30, 50, 70, 90, and 100%) and dried with a critical point drier (Leica EM CPD300, Germany). The samples were mounted and sputter-coated with platinum and imaged using a NOVA NanoSEM 450 scanning electron microscope.

### Membrane integrity assay
PI assays were used to evaluate the ability of cAMPs to disrupt cytoplasmic membrane integrity. *A. baumannii* ATCC 19606 was cultured to an $OD_{600}$ of 0.5, centrifuged (10,000 × g, 5 min, 4 °C), washed, and resuspended in 10 mM PBS (pH 7.0) to a working concentration ($OD_{600}$ = 0.1). 100 $\mu$l bacterial suspension was mixed with 100 $\mu$l of cAMP solution in a 96-well plate to reach a final peptide concentration of 1 × MIC and 10 × MIC for 1 h. The negative control was treated with buffer alone. Subsequently, PI solution (40710ES03, Yeasen, Shanghai, China) was added at a final concentration of 20 $\mu$M and incubated at 37 °C for 30 min in the dark. Fluorescence was measured with the excitation wavelength at 535 nm and emission wavelength at 615 nm.

### Longitudinal resistance assay
To assess the resistance development during serial passage, *E. coli* ATCC 25922, *A. baumannii* ATCC 19606, *A. baumannii* BAA-1605, and *E. faecium* BM4105 were cultured in MHB at 37 °C overnight. Subsequently, 2 $\mu$l of this culture was used to inoculate 200 $\mu$l of fresh MHB containing cAMP_81 at a concentration of 0.5 × MIC. This process was repeated every 24 h: inoculating 200 $\mu$l of fresh MHB containing cAMP_81 at a concentration of 0.5 × MIC with the previous culture, at a 1% inoculum ratio. This cycle was continued for 30 days, and MIC assays were performed on the resultant bacteria at 10 day intervals.

### Statistics & Reproducibility
All experiments were independently repeated at least two times with consistent results to confirm reproducibility. No statistical method was used to predetermine sample size. Sample sizes for each experiment were determined based on previous published studies in the field, ensuring sufficient statistical power. No metagenomic datasets were excluded except those failing standard quality-control criteria. No experimental data were excluded from the analyses. The experiments were not randomized. The study did not include any interventions and thus the conventional blinding was not applicable to this study.

### Reporting summary
Further information on research design is available in the Nature Portfolio Reporting Summary linked to this article.

## Data availability
All 74,999 MAGs generated in this study, together with 83 in-house isolate genomes from deep-sea, the non-redundant gene sets from assembled contigs and genomes, and 163,693 BGCs, have been deposited in the China National GeneBank DataBase (CNGBdb) with accession number CNP0007106. The accession IDs of publicly available bacterial and archaeal reference genomes from NCBI genome database are provided in Supplementary Data 2. The referenced representative genomes used in this study, including 113,104 from GTDB release R220, 22,732 from GEM[47], 24,195 from GOMC[19], and 957 from Tara Ocean[46], are available at https://gtdb.ecogenomic.org/, https://portal.nersc.gov/GEM/genomes/, https://db.cngb.org/maya/datasets/MDB0000002, and https://merenlab.org/data/tara-oceans-mags/, respectively. The 4472 representative genomes from UHGG v2.0[108] used in this study are available at https://www.ebi.ac.uk/metagenomics/genome-catalogues/human-gut-v2-0-2. All additional data supporting the findings of this study are provided within the main text, Supplementary Information files, or via the provided repositories. Source data are provided with this paper.

## Code availability
The trained model weights and corresponding datasets are now publicly available at Zenodo (https://zenodo.org/records/17613552). The inference scripts enabling reproduction of the model results are publicly available on our GitHub repository (https://github.com/BGI-METAI/Metagenome-AI), together with Python Jupyter notebooks for

creating the figures and tables with model results from this manuscript.

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

## Acknowledgements

This study was supported by the Key Research and Development Program of Hainan Province (grant No. ZDYF2024SHFZ046 to Haixin Chen), the Project of Sanya Yazhou Bay Science and Technology City (grant No. SKJC-2024-01-002 to X.F., SKJC-2024-01-001 to Haixin Chen, and SCKJ-JYRC-2023-41 to P.J.), Hainan Provincial Natural Science Foundation of China (grant No. 826QN0909 to P.J.), and Hainan Yazhou Bay Seed Lab (grant No. JBGS B23YQ2003 to Z.Y.). Computations in this study were supported by the High-performance Computing Platform of YaZhou Bay Science and Technology City Advanced Computing Center. The authors thank Kui Zhu (China Agricultural University) for kindly donating the strains *S. aureus* ATCC 29213, *E. faecalis* ATCC 29212, *E. faecalis* VRE10, and *E. coli* ATCC 25922 for AMP antibacterial tests; Cong Shen (Guangdong Provincial Hospital of Chinese Medicine) for providing bacterial strains *E. faecium* BM4105, *P. aeruginosa* ATCC 27853, and *P. aeruginosa* PAO1 for AMP antibacterial tests; and our colleague Jun Wang for his help with the bioinformatic analysis. The authors thank the China National GeneBank, BGI Research, Shenzhen 518120, China, for their support. The authors thank ChatGPT-4o mini (OpenAI) for assistance with language editing and phrasing. Supplementary Fig. 1 was created with Biorender.com.

## Author contributions

H.C. (Haixin Chen), P.Y., C.X., Z.Y., and P.J. conceived and supervised the study. P.J., Z.L., F.W., and L.L. collected the data and contributed to formal analyses. P.J. and Z.L. conducted bioinformatic analyses and data visualization. P.Y. and V.K. conceived the development of the MAI framework. V.K. and N.M. developed the MAI algorithm. C.P. and N.S. evaluated the MAI framework and published AMP models. M.H., Y.Z., Y.X.L., and J.H.L. provided computational resources and bioinformatic analysis support. C.X., J.S., and P.J. designed the experimental validation. C.X., X.F., and R.C. provided experimental platforms and resources. Y.L. isolated and sequenced microbial strains from cold seeps. J.S., X.L., L.W., S.W., H.C. (Haixian Cheng), J.N.L., and Y.J. performed all in vitro experiments. J.S. and P.J. analyzed and interpreted the experimental results. P.J., V.K., and Z.L. drafted the manuscript. All authors reviewed and approved the final version of the manuscript.

## Competing interests

The authors declare no competing interests.

## Additional information

**Puzi Jiang** [1,2,8], **Zhengjiao Liang**[1,2,8], **Vladimir Kovacevic** [3,8], **Jingya Shi**[1,2,8], **Nikola Milicevic**[3], **Feng Wang**[1], **Lin Liu**[4], **Yue Liu** [1,2], **Yunjiang Jiang**[1,2], **Mo Han** [1,2], **Xiaonan Lin**[1], **Časlav Petronić**[3], **Nikola Stanojevic**[3], **Lingqin Wang**[5], **Suwan Wang**[5], **Haixian Cheng**[5], **Jiani Li**[5], **Rouxi Chen**[1,2], **Yong Zhang** [4], **Yuxiang Li** [4], **Junhua Li** [3], **Xiaodong Fang**[1,2,6], **Zhen Yue** [1,2] ✉, **Chuang Xue** [5,7] ✉, **Peng Yin** [4] ✉ & **Haixin Chen** [1,2] ✉

[1]BGI Research, Sanya, China. [2]Hainan Technology Innovation Center for Marine Biological Resources Utilization (Preparatory Period), BGI Research, Sanya, China. [3]BGI Research, Belgrade, Serbia. [4]BGI Research, Wuhan, China. [5]MOE Key Laboratory of Bio-Intelligent Manufacturing, State Key Laboratory of Fine Chemicals, Frontiers Science Centre for Smart Materials Oriented Chemical Engineering, School of Bioengineering, Dalian University of Technology, Dalian, China. [6]BGI Research, Shenzhen, China. [7]Ningbo Institute of Dalian University of Technology, Ningbo, China. [8]These authors contributed equally: Puzi Jiang, Zhengjiao Liang, Vladimir Kovacevic, Jingya Shi. ✉e-mail: yuezhen@genomics.cn; xue.1@dlut.edu.cn; yinpeng@genomics.cn; chenhaixin@genomics.cn

