## [Transparent Peer Review file · Nature Communications]

The Extreme Environment Microbiome Catalog (EEMC): A Global Resource for Microbial Diversity and Antimicrobial Discovery

Corresponding Author: Mr Haixin Chen

Version 0:

Reviewer comments:

Reviewer #1

(Remarks to the Author)

This manuscript introduces the Extreme Environment Microbiome Catalog (EEMC), assembled from 2,293 metagenomes and 3,214 isolate genomes collected across seven extreme-habitat types. The authors reconstruct 78,213 archaeal and bacterial genomes that cluster into 32,715 species-level OTUs, an estimated 63% of which have no match to existing taxonomies. They further assemble ~4 billion non-redundant genes and 163,693 biosynthetic gene clusters (BGCs), of which 58.7% fall outside the current BiG-SLiCE reference space. Finally, they train several protein-language-model (pLLM) classifiers (ESM2-3B, ESM-3, PTrans) to predict antimicrobial activity and mammalian toxicity of RiPP core peptides, yielding 3,032 candidate non-toxic AMPs. One hundred peptides were then synthesized; 84 inhibited at least one pathogen, and six displayed sub-10 μM MICs against multidrug-resistant Gram-negative strains. The authors position EEMC as a resource for microbial ecology and natural-product discovery. Overall, the catalogue is valuable, and the integration of pLLM-guided AMP discovery is timely. However, several major issues need to be addressed (see below) prior to any further consideration. In its current form the work is too preliminary for Nature Communications standards.

Specific comments:

1. Genomes as low as 50% completeness and 10% contamination (mean completeness $\approx 77\%$) are included. Such relaxed criteria can inflate species counts and mis-assign BGCs or other functional elements.
2. The 2,293 metagenomes span 2010–2024 and use varied extraction and sequencing protocols. Habitat-driven gene or BGC specificity is interpreted without demonstrating that study or batch effects are negligible.
3. Of the 163,693 BGCs, > 50% are labeled “novel”, yet many originate from medium-quality MAGs and are fragmented. The effect of cluster fragmentation on distance-based novelty scoring should be evaluated.
4. Reported accuracies of up to 96% for AMP and toxicity prediction are given without showing how sequence-identity overlap between public training datasets and EEMC peptides was removed, risking over-optimistic performance estimates.
5. No assays for serum stability, resistance evolution, or in vivo efficacy are provided, which are standard in the field, further leaving translational claims unsupported by data. I really think testing the peptides in at least an animal infection model is a major issue and critical here.
6. The authors should provide all accession numbers/DOIs for raw assemblies, gene catalogues, trained model weights, and inference scripts to enable reproducibility of the work.
7. The discussion should benchmark the six potent AMPs against published examples (e.g., prevotellin-2) and compare the MAI framework with prior microbiome-mining pipelines for peptide discovery. Moreover, the authors do not discuss recent mining efforts of archaeal proteomes for bioactive antibiotic peptides.

Minor comments:

1. The ecological or clinical relevance of millions of putative virulence factors is speculative. I suggest highlighting only experimentally validated or high-confidence homologues.
2. Please check the manuscript for typos – e.g. “environment” (line 71), “archael” (line 74).
3. The authors mention the lack of generalizable models, but this is not entirely accurate (e.g., APEX).
4. References. Check for issues. E.g., update citation 37 (Macrel) year (should be 2020, not 2010).

(Remarks on code availability)

Reviewer #2

(Remarks to the Author)

The manuscript by Jiang and co-authors contributes a significant resource to the scientific community with the generation of the Extreme Environment Microbiome Catalog (EEMC) as well as making an important contribution to the field of antimicrobial peptides and genome mining for these bioactive compounds. The authors collected publicly available (meta)genomes from extreme environments, assembled MAGs from these, then analyzed them (including some pure isolate genomes from their own collection) for the presence of BGCs, with an emphasis on RiPPs. Separately they created a curated database of antimicrobial peptides (active against either Gram + or Gram – bacteria or with broad spectrum), but non-toxic, pulled from available databases. This was used as training and validation data for their custom machine learning model (Metagenome-AI) used to develop classifiers capable of detecting peptides with these desirable parameters from a pool of uncharacterized RiPPs, mined from the MAGs they reconstructed. Next, they selected 100 of the peptides predicted to be antimicrobial to be chemically synthesized. These peptides were assayed for their antimicrobial activity against a range of Gram + and Gram - bacteria. These experiments revealed that a high proportion of the peptides selected (84/100) demonstrated antimicrobial activity and seven of these were demonstrated with a CC50 higher than their MIC suggesting that they have therapeutic potential.

General Comments / Corrections to be addressed:

1. In lines 162–164, the authors suggest that microorganisms in extreme environments may gain a competitive advantage due to the widespread presence of secondary metabolite pathways. While this is a well-established ecological principle across diverse habitats, the current analysis does not include comparative data from non-extreme environments, making it difficult to determine whether this trait is uniquely enriched or more pronounced in the sampled extremes. The statement may unintentionally imply such uniqueness, and a brief clarification would help ensure the intended nuance is preserved. Alternatively, provide references to support the idea that this is particularly enriched in extreme environments, if that is what the authors intended.
2. The paragraph spanning lines 174–189 places considerable emphasis on the potential threat to human health posed by virulence factors. However, this framing seems at odds with several findings that highlight the novelty and niche specificity of the microbial genomes and functions: (i) the remarkably limited sharing of unigenes across environments; (ii) low percentage similarity to the GEM and UHGG databases; (iii) biosynthetic gene cluster (BGC) analysis indicating a high proportion of habitat-specific GCFs and GCCs; and (iv) the identification of highly novel virulence factors. These results support the authors' conclusion that strong environmental selective pressures drive the evolution of microbial lineages toward peptides with specialized ecological roles. In this context, virulence appears to be highly adapted to specific environmental conditions and may not be directly translatable to human pathogenicity. A more balanced framing would better reflect the ecological significance of these findings. However, the rationale for prioritising this analysis over other metabolic traits is not clear. As presented, it does not substantially advance the study's main focus. To strengthen the manuscript's coherence, the authors might consider either omitting this section or, alternatively, expanding the scope to include comparable analyses of other functional traits. This would provide a more balanced and contextually grounded assessment of microbial adaptation mechanisms in extreme environments.
3. In line 182, the authors report up to 1,191 virulence factors within a single genome. This figure appears unusually high and may warrant further context. Could the authors comment on whether such a number is typical, or whether it might reflect an overestimation due to annotation thresholds or database limitations? It would be helpful to reference comparable cases in the literature where similarly high counts of virulence factors have been reported, to support the plausibility of this finding.
4. The authors provided access to the MAGs, as well as the software pipeline they created, but not the final curated list of peptides used as training the dataset nor the final trained model. If these are not to be patented, it would be a fantastic resource for the rest of the microbial and peptide community in particular. Could the authors comment on why these were not made available?
5. What is the distribution of the 108 cAMPs between MAGs and isolates?
6. While acknowledging the cost and logistical constraints, it would have strengthened the study if the authors had also synthesized and assayed a comparable number of peptides from the EEMC predicted to be non-antimicrobial. This would have provided a more balanced assessment of the predictive model's specificity, demonstrating its ability not only to identify true positives but also to exclude false positives.
7. Many RiPPs derive their bioactivity from post-translational modifications (PTMs) such as glycosylation and stereoselective modifications. Given that 96 of the 100 synthesized cAMPs were classified as PTM peptides (lassopeptides and class II lanthipeptides), did the synthetic synthesis incorporate these PTMs and what structural validations were performed beyond mass confirmation and purity? This could be a critical gap, as PTMs, especially those conferring conformational rigidity like disulfide bridges or macrocyclization, are essential for the biological activity of RiPPs. Given that DeepRiPP was used to predict core peptides independently of genomic context, and that most synthesized peptides were lassopeptides and lanthipeptides, it is likely that the predicted sequences represent only the core scaffold, not the full PTM

landscape. Therefore, the authors should clarify whether synthetic strategies attempted to mimic native PTMs (e.g., via chemical cyclization or enzymatic modification), and whether stereochemistry was verified (e.g., via NMR or circular dichroism). Without such validation, the observed cytotoxicity and antimicrobial activity may not fully reflect the native peptide's properties, potentially limiting translational relevance. This does not detract from the impressive success rate; however, it may help explain why the observed activities predominantly reflect similar modes of action and lack novelty in terms of distinct scaffolds or mechanistic diversity.

8. Building on the previous point, the current AMP prediction framework relies predominantly on sequence embeddings without incorporating structural or physicochemical modeling. This approach may overlook context-dependent activity and risks misclassifying peptides with atypical conformations or noncanonical features. Could the authors expand on how the PTMs were accommodated in their ML training? Furthermore, the authors should acknowledge that the predicted peptides may differ significantly from their naturally occurring counterparts in terms of folding, stability, and bioactivity. To clarify the scope and applicability of the models, it is important to specify whether the training datasets were derived from synthetically produced or naturally sourced peptides, and whether the associated activity data reflect in vitro assays or native biological contexts. Furthermore, the authors are encouraged to discuss how structure-based modeling and molecular dynamics simulations could be integrated to complement sequence-based predictions, thereby enhancing the reliability and interpretability of AMP discovery.

Minor Corrections

Ln67: delete "s" from sequences, ie should be "sequence datasets"

Ln147: add a space between "plateauing" and (Supplementary....)

Ln160: metabolic regulation

Ln239: niche-specificity (not niche-specific)

Line 265: Linear azole-containing peptides

Supplementary figure6 – figure legend, two commas after e-f and are the figures in e and f duplicates (f does not refer to genera and the curves are identical)?

(Remarks on code availability)

Yes, I could install and run the code and the authors provide a small dataset to test the program with. Because the authors don't make their database of peptides available, it was not possible to replicate the results in the manuscript. We would strongly suggest that the authors consider making the peptide database available, as well as the trained model which could be a valuable resource for the community.

Reviewer #3

(Remarks to the Author)

(Remarks on code availability)

Yes, I could install and run the code and the authors provide a small dataset to test the program with. Because the authors don't make their database of peptides available, it was not possible to replicate the results in the manuscript. We would strongly suggest that the authors consider making the peptide database available, as well as the trained model which could be a valuable resource for the community.

Version 1:

Reviewer comments:

Reviewer #1

(Remarks to the Author)

The authors have addressed most of my prior comments.

(Remarks on code availability)

Reviewer #2

(Remarks to the Author)

It is evident that the authors have undertaken an immense amount of work to address the reviewers' concerns, and their efforts in expanding the experimental validation and refining the genome-quality thresholds have significantly improved the technical rigor of the study. Thank you.

The study approach creates a sequence-function correlation that completely ignores the natural biological context of these peptides. Moreover, it leaves one questioning the rationale for focusing on RiPPs specifically. Theoretically, the authors

could have screened the entire sequence space for random peptide libraries and identified active sequence combinations through their model with similar results. By bypassing the very modifications that define RiPPs, the biological relevance of using these specific BGCs as a starting point is significantly diminished. Is this something the authors have ever considered?

The absence of PTMs likely narrowed the chemical space explored and resulted in a convergence of observed modes of action. While the success rate in identifying active peptides (84 out of 100) is technically impressive, the concerns pointed out detract from the study's ability to reveal truly novel or biologically "native" chemical diversity.

(Remarks on code availability)

Reviewer #3

(Remarks to the Author)

(Remarks on code availability)

Response to the reviewers

Reviewer #1 (Remarks to the Author):

This manuscript introduces the Extreme Environment Microbiome Catalog (EEMC), assembled from 2,293 metagenomes and 3,214 isolate genomes collected across seven extreme-habitat types. The authors reconstruct 78,213 archaeal and bacterial genomes that cluster into 32,715 species-level OTUs, an estimated 63% of which have no match to existing taxonomies. They further assemble ~4 billion non-redundant genes and 163,693 biosynthetic gene clusters (BGCs), of which 58.7% fall outside the current BiG-SLiCE reference space. Finally, they train several protein-language-model (pLLM) classifiers (ESM2-3B, ESM-3, PTrans) to predict antimicrobial activity and mammalian toxicity of RiPP core peptides, yielding 3,032 candidate non-toxic AMPs. One hundred peptides were then synthesized; 84 inhibited at least one pathogen, and six displayed sub-10 μ M MICs against multidrug-resistant Gram-negative strains. The authors position EEMC as a resource for microbial ecology and natural-product discovery. Overall, the catalogue is valuable, and the integration of pLLM-guided AMP discovery is timely. However, several major issues need to be addressed (see below) prior to any further consideration. In its current form the work is too preliminary for Nature Communications standards.

Response:

Thanks for the thorough evaluation and constructive comments. We appreciate the recognition of the value of the EEMC and the integration of pLLM-guided AMP discovery. We also acknowledge your concerns regarding aspects of the current analysis. The feedback is highly valuable, and we have carefully revised the manuscript to address all major points raised.

In brief, we performed additional analyses to assess the quality of genomes, genes, and BGCs of the EEMC, expanded the experimental evaluation of potent cAMPs, compared the MAI framework against a previously published microbiome-mining AMP predictor, and corrected minor issues throughout the text. We have also made all curated datasets and trained models publicly accessible to support transparency and reproducibility. We hope that these revisions substantially improve the rigor and clarity of the manuscript and address the reviewer's concerns.

Specific comments:

1. Genomes as low as 50% completeness and 10% contamination (mean completeness \approx 77 %) are included. Such relaxed criteria can inflate species counts and mis-assign BGCs or other functional elements.

Response:

We agree that genome quality is crucial for reliable downstream inferences. Our inclusion thresholds (\geq 50% completeness, $<$ 10% contamination) meet the medium-quality standard of the Minimum Information about a Metagenome-Assembled Genome (MIMAG)¹, which is widely used in large-scale metagenome-assembled genome (MAG) catalogues spanning diverse environments, such as the Tibetan Plateau Microbiome Catalog (TPMC), Soil Microbial Genome Catalog (SMAG),

and Ocean Microbiome Database (OMD)²⁻⁴. The mean completeness of EEMC genomes (76.84%) is comparable to these catalogues (78.05% in TPMC, 76.07% in SMAG, and 71.76% in OMD). Although the catalogues shown in Fig. 1d applied different filtering strategies (e.g., > 70% completeness or > 2 Mbp length in TARA MAGs), the mean completeness of EEMC genomes still falls within the range of these catalogues (69.97% in TARA⁵, 82.33% in GOMC⁶, and 83% in GEM⁷). Furthermore, the proportion of EEMC genomes with a quality score (QS, calculated as completeness – 5 × contamination) less than 50 is 25.9%, which is within the range observed in these catalogues (22.68% in TPMC, 33.42% in SMAG, and 20.85% in OMD). Together, these results indicate that our dataset falls well within the accepted quality spectrum of recently published large-scale metagenomic resources.

To further address your concern, we stratified genomes by QS = 50 and re-evaluated species-level novelty. The 57,946 higher-quality genomes (QS > 50) were clustered into 24,636 representative species, of which 13,781 (55.94%) were classified as novel, referring to the GTDB r220 database⁸, slightly lower than the value for the full dataset (63.00%) but consistent in general (**Response Table 1**). The 20,267 lower-quality genomes (QS ≤ 50) were clustered into 14,033 representative species, of which 10,294 (73.36%) were classified as novel (**Response Table 1**). Importantly, novelty relative to the public genome catalogues remains almost unchanged across QS-stratified genome sets (**Response Table 1**), indicating that species-level novelty is robust to genome-quality filtering.

In the previous BGC analyses, to minimize the potential effects of genome fragmentation and mis-assignment, antiSMASH was executed with the parameter “--minlength 5000” to restrict BGC predictions to contigs ≥ 5 kb. This yielded 163,693 BGCs across the 78,213 genomes, of which 19,735 (12.06%) originated from genomes with QS ≤ 50. We evaluated the completeness and novelty of BGCs derived from QS-stratified genome sets. As expected, the proportion of complete BGCs was lower in genomes with QS ≤ 50 (2.90%) than in the full genome set (19.97%) and in higher-quality genomes (QS > 50) (22.31%) (**Supplementary Fig. 5c**). Despite this, the class-level composition of BGCs remained consistent across the genome sets (**Supplementary Fig. 5d**). Novelty among BGCs from QS ≤ 50 genomes (47.37% of GCFs and 44.65% of GCCs) remains high (**Response Fig. 1**), indicating that lower-quality genomes still encode a substantial amount of biosynthetic novelty.

For the gene set analyses, nearly 4 billion non-redundant genes were predicted directly from metagenomic contigs, aiming to comprehensively capture the potential genetic repertoire of the EEMC without being constrained by genome-quality thresholds.

For the genome-derived gene analyses, we retained only MAG contigs ≥ 1 kb to mitigate issues related to assembly fragmentation. Of 203,220,315 genes predicted from 78,213 EEMC genomes, 47,319,159 genes (23.28%) originated from genomes with QS ≤ 50. As expected, these genes showed higher incompleteness (39.72%) compared to QS > 50 genomes (17.87%) (**Response Fig. 2a**). To address your concern, we assessed the novelty of the gene sets derived from QS-stratified genomes by clustering these genes with those from the GEM and GOMC genome catalogs. Notably, 83.30% of gene clusters originating from QS ≤ 50 genomes were classified as novel, slightly higher than the 81.3% observed for clusters from higher-quality genomes (**Response Fig. 2b-c**). Although lower-quality genomes contain a higher proportion of incomplete genes, they contribute substantially to the overall genetic diversity and novelty of the EEMC, supporting their inclusion in our analyses.

Taken together, these results indicate that the inclusion of a limited number of lower-quality genomes does not substantially affect our conclusions regarding species-level diversity, novelty, BGC composition, or gene novelty in the genome-derived gene catalog. Instead, these genomes contribute genomic and genetic diversity and novelty, supporting their inclusion in this study.

Reference database	Novelty of QS \leq 50 genome set (%)	Novelty of QS $>$ 50 genome set (%)	Novelty of full genome set (%)
GTDB r220	73.36	55.94	63.00
TARA	99.86	99.79	99.85
GEM	96.18	94.55	95.56
GOMC	92.14	85.70	88.42

Response Table 1. The novelty of the full and QS-stratified genome sets.

Supplementary Fig. 5. **c**, Completeness of BGCs derived from the full genome and QS-stratified genome sets of EEMC. **d**, Counts of BGC classes across the full and QS-stratified genome sets (left panel), and relative proportions of BGC classes in the full, complete, partial, and lower-QS genome-derived BGC datasets (right panel).

Response Fig. 1. Novelty of GCCs and GCFs in BGCs derived from lower-QS genome set ($QS \leq 50$).

Response Fig. 2. a, Completeness of genes derived from the full, $QS > 50$ and $QS \leq 50$ genome sets of EEMC. **b,** Number of gene clusters shared with GEM or GOMC in the full, $QS > 50$ and $QS \leq 50$ genome sets. **c,** Proportion of gene clusters shared with GEM or GOMC in the full, $QS > 50$ and $QS \leq 50$ genome sets.

Specific revisions:

We have conducted QS-stratified analyses to support the conclusions of species-level diversity, novelty, BGC composition, and gene novelty in the genome-derived gene catalog.

Accordingly, we have added the following content to the Results section of the revised manuscript: (i) explicit reporting of genome quality scores ($QS = \text{completeness} - 5 \times \text{contamination}$) and their distribution across all genomes (L. 102, L. 584-586), (ii) a breakdown of the quality of genomes assigned to putative novel species (L. 119-121), and (iii) additional analyses comparing BGC yield, completeness, and class-level composition across the full and QS-stratified datasets (L. 233-241). New supplementary figures (Supplementary Fig. 5c-d) have been included to illustrate the results.

The newly added contents read as follows:

1. L. 102: “57,946 exhibited a quality score (QS) above 50 (calculated as completeness - (5×contamination))”

2. L. 584-586: “Retained genomes were further assigned a QS, calculated as completeness - 5 × contamination, following the implementation in the Tibetan Plateau Microbial Catalog (TPMC)¹⁶.”

3. L. 119-121: “Among these 20,610 putative novel species, 13,785 had a QS greater than 50, and 1,483 met the high-quality genome criteria (Supplementary Data 2).”

4. L. 233-241: “We first used antiSMASH (version 7.0.1) on the 78,213 microbial genomes to identify 163,693 BGCs, of which 143,958 (87.94%) originated from genomes with $QS > 50$. As expected, the proportion of complete BGCs was lower in genomes with $QS \leq 50$ (2.90%) than in the full genome set (19.97%) and higher-quality genomes ($QS > 50$) (22.31%) (Supplementary Fig. 5c). In addition, only 14.63% of all partial BGCs (19,161 out of 131,001) came from genomes with $QS \leq 50$ (Supplementary Fig. 5c), indicating that fragmented BGCs are not predominantly contributed by lower-QS genomes. The 163,693 BGCs were further categorized into eight classes, with lengths ranging from 4,691 to 288,454 bp (Supplementary Data 6). The distribution of class-level composition remained consistent across the full and QS-stratified genome datasets (Supplementary Fig. 5d). These results indicate that the inclusion of a limited number of lower-quality genomes does not materially affect our conclusions regarding overall BGC diversity and distribution patterns.”

2. The 2,293 metagenomes span 2010–2024 and use varied extraction and sequencing protocols. Habitat-driven gene or BGC specificity is interpreted without demonstrating that study or batch effects are negligible.

Response:

Thank you for your insightful comments. We acknowledge that batch effects introduced by varied extraction and sequencing protocols are a common phenomenon in large-scale metagenomic studies. To prevent potential misinterpretation, we have removed statements and hypotheses regarding the habitat-specificity of species, genes, and BGCs across extreme environments. We have also revised the corresponding descriptions of species, genes, and BGCs distributions to focus on observed patterns without over-interpreting habitat specificity.

Specific revisions:

We have removed statements implying habitat-driven specificity of species, genes, and BGCs, including L. 137-159, L. 190-191, L. 212-227, L. 260-262, L. 289-294, L. 306-308, and L. 442-443. The corresponding sections in the Results have been revised to describe distributions without

inferring causality or ecological specificity (L. 123-127, L. 132-137, L. 185-188, L. 294-296, L. 308-309). These revisions ensure a more cautious interpretation given the potential influence of study- or protocol-related batch effects.

The newly revised contents read as follows:

1. **L. 123-127:** “Moreover, the majority of unclassified OTUs (uOTUs) were singleton genomes (67.62%), markedly exceeding the proportion observed in known OTUs (kOTUs) (54.41%) (Supplementary Fig. 2a). We further found that the proportion of uOTUs within the dominant phyla ($\leq 80\%$), such as Gemmatimonadota and Planctomycetota, was lower than that in rare phyla ($> 85\%$), including Bdellovibrionota and Hydrogenedentota (Fig. 1e, Supplementary Fig. 2b). This pattern indicates the critical contribution of the EEMC in recovering rare species of extreme environment microbiomes.”

2. **L. 132-137:** “Distinct phylum-level patterns of uOTUs were observed across extreme environments. For instance, Pseudomonadota, Bacteroidota, and Actinomycetota were highly represented among uOTUs from the cryosphere, while Halobacteriota were frequent among those from hypersaline regions. Actinomycetota contributed substantially to uOTUs in hyperarid regions, and Chloroflexota and Thermoproteota were notable among uOTUs from terrestrial geothermal habitats. Overall, these observations highlight the broad phylogenetic diversity of uOTUs captured by the EEMC across extreme environments.”

3. **L. 185-188:** “microorganisms from extreme environments likely employ chemical strategies for competition and interaction, an ecological mechanism broadly observed across diverse microbial ecosystems^{2,11,61}. Given the considerable number of novel genes and biosynthetic features uncovered in the EEMC, the microbes in these habitats may therefore represent valuable reservoirs of novel natural products.”

4. **L. 294-296:** “Similar environment-associated patterns of biosynthetic diversity have also been observed in other large-scale microbiome studies, such as fermented food and human microbiomes^{63,64}.”

5. **L. 308-309:** “These observations suggest that RiPPs may be linked to ecological adaptations and host interactions²⁸, though sampling and sequencing biases may also play a role⁶⁵.”

Supplementary Fig. 2. Singleton rates and taxonomic patterns of uOTUs and kOTUs. **a**, The proportion of singleton genomes within uOTUs and kOTUs. **b**, The OTU counts and composition

in phyla with ≥ 10 OTUs and $\geq 80\%$ uOTUs.

Fig. 1. e, The number and proportion of kOTUs and uOTUs across the top 20 phyla. **f**, The number and top 10 phyla composition of uOTUs across extreme environments.

3. Of the 163,693 BGCs, > 50% are labeled “novel”, yet many originate from medium-quality MAGs and are fragmented. The effect of cluster fragmentation on distance-based novelty scoring should be evaluated.

Response:

Thanks for this important observation. We agree that genome quality and BGC completeness are critical for reliable novelty estimation. To mitigate the impact of fragmented assemblies, BGCs were predicted using antiSMASH with the parameter “--minlength 5000”, which restricts BGC prediction to contigs ≥ 5 kb. This stringent filtering yielded 163,693 BGCs across the 78,213 genomes, of which 19,735 (12.06%) originated from genomes with $QS \leq 50$. As expected, the proportion of complete BGCs was lower in genomes with $QS \leq 50$ (2.90%) than in the full genome set (19.97%) and higher-quality genomes ($QS > 50$) (22.31%) (**Supplementary Fig. 5c**). In addition, only 14.63% of all partial BGCs (19,161 out of 131,001) came from genomes with $QS \leq 50$, indicating that fragmented BGCs are not predominantly contributed by lower-QS genomes. We then evaluated the effect of genome quality on novelty scoring. The class-level composition of BGCs remained consistent across the genome sets (**Supplementary Fig. 5d**). Novelty among BGCs from $QS \leq 50$ genomes (47.37% of GCFs and 44.65% of GCCs) remains high (**Response Fig. 1**), indicating that lower-quality genomes still encode non-negligible biosynthetic novelty.

To evaluate the effect of cluster fragmentation on novelty scoring, we stratified the BGC dataset according to completeness and assessed the novelty of the two BGC subsets relative to public databases. The 131,001 incomplete BGCs and 32,692 complete BGCs were clustered into 55,184 and 16,372 gene cluster families (GCFs, 0.3 distance threshold), and further into 1,755 and 219 gene cluster clans (GCCs, 0.7 distance threshold), respectively. To avoid sampling bias, we dereplicated by retaining only the longest BGC per GCF per species, yielding 82,526 and 29,907 representative incomplete and complete BGCs, respectively.

Following the BiG-FAM framework, query BGCs with distances ≤ 900 were defined as “matched”, while those > 900 were classified as “novel”. GCFs with $< 20\%$ matched members and GCCs with $< 40\%$ matched members were designated as novel. Based on these criteria, 77,264 (58.98%) and 25,880 (79.17%) novel BGCs were identified from the incomplete and complete BGC sets, respectively (**Supplementary Data 6**). In the complete BGC set, 12,986 (79.32%) GCFs and 170 (77.63%) GCCs were classified as novel, while in the incomplete BGC set, 30,010 (54.38%)

GCFs and 1,148 (65.41%) GCCs were identified as novel. The proportions of novel clusters in both complete and incomplete BGCs were consistent with those in the full BGC dataset (58.68% novel GCFs and 70.16% novel GCCs). Additionally, the overall class composition and novelty proportions within each BGC class remained largely consistent across the full BGCs and the two completeness-stratified subsets (**Fig. 3c, Supplementary Fig. 5d-e**).

Together, these results indicate that the cluster fragmentation and genome quality have only a modest influence on distance-based novelty scoring, and the high novelty proportion observed in EEMC reflects genuine biosynthetic divergence rather than assembly artifacts.

Supplementary Fig. 5. c, Completeness of BGCs derived from the full genome and QS-stratified genome sets of EEMC. **d**, Counts of BGC classes across the full and QS-stratified genome sets (left panel), and relative proportions of BGC classes in the full, complete, partial, and lower-QS genome-derived BGC datasets (right panel). **e**, Novelty of GCCs and GCFs in partial and complete BGC sets.

Response Fig. 1. Novelty of GCCs and GCFs in BGCs derived from lower-QS genome set ($QS \leq 50$).

Fig. 3. c, The 163,693 BGCs were clustered into 64,733 GCFs and 2,178 GCCs, with their novelty and the total count across different types indicated.

Specific revisions:

We have added QS-stratified and completeness-stratified analyses to evaluate the effect of genome quality and cluster fragmentation on distance-based novelty scoring. These results have been incorporated into the Results section (L. 238-241, L. 268-272) and accompanied by new supplementary figures (Supplementary Fig. 5d–e) in the revised manuscript.

The newly added contents read as follows:

1. L. 238-241: “The distribution of class-level composition remained consistent across the full and QS-stratified genome datasets (Supplementary Fig. 5d). These results indicate that the inclusion of a limited number of lower-quality genomes does not materially affect our conclusions regarding overall BGC diversity and distribution patterns.”

2. L. 268-272: “The overall composition of BGC classes and the novelty proportions within each BGC class were largely consistent between the full BGCs and the two completeness-stratified subsets (Fig. 3c, Supplementary Fig. 5d-e), indicating that cluster fragmentation had only a modest impact on distance-based novelty scoring, and that the high proportion of novel BGCs in the EEMC likely reflects genuine biosynthetic divergence rather than assembly artifacts.”

4. Reported accuracies of up to 96% for AMP and toxicity prediction are given without showing how sequence-identity overlap between public training datasets and EEMC peptides was removed, risking over-optimistic performance estimates.

Response:

We understand your concern regarding sequence-similarity dependent effects. We clarified that both the training and test datasets for the AMP and toxicity prediction models were derived exclusively from publicly available peptide databases (including APD3, CAMP, DRAMP, DBAASP, dbAMP, LAMP2, BaAMP, Hemolytik, and ToxinPred) (L. 654-655, L. 674). The EEMC-derived RiPP core peptides were not used in any stage of model training, validation, or testing, and were applied solely for inference once the models were fully trained. In addition, we have revised Fig. 4a in the manuscript to clarify the MAI framework and prevent potential misinterpretation.

To further confirm the data independence, we performed a pairwise Needleman–Wunsch (NW) similarity analysis between the RiPP core peptides derived from the EEMC and peptides in training datasets. The distribution of similarity scores showed that most RiPP sequences share low similarity with the training data, indicating that the observed accuracies of prediction were primarily determined by their performance on novel and unseen sequences (Supplementary Fig. 7b-c). We found 14 RiPP peptides that are identical to entries in the AMP training dataset and 1 peptide identical to an entry in the toxicity training dataset (Supplementary Table 15). These duplicated peptides correspond to sequences previously reported as AMPs or toxic peptide. Importantly, we excluded these duplicated peptides from experimental synthesis and validation, thereby preserving the novelty of the candidate non-toxic AMPs tested. Given the very small number of duplicated sequences relative to the full RiPP set, the independence between the training sequences and the EEMC-derived peptides remains largely preserved, and there is no evidence of systematic data leakage.

Supplementary Fig. 7. b-c, Distribution of highest similarity between 11,379 RiPP core peptides to peptides of the global AMP training dataset (b) and toxicity training dataset (c).

Fig. 4. a, Data collection and deep learning architecture of the pLLM-based models used for predicting antimicrobial activity and toxicity. The AMP global models were trained on 25,004 AMP sequences and 30,205 non-AMP protein sequences. The AMPs with anti-G+ or anti-G- label and non-AMPs were further used to train anti-G+ AMP models and anti-G- AMP models, respectively. The toxicity models were trained on 6,318 toxic and 6,643 non-toxic protein sequences. The trained pLLM-based models were used to predict the antimicrobial activity and toxicity of 11,379 RiPPs.

Specific revisions:

We have added a NW similarity analysis (L. 706-709) and confirmed the independence between RiPP core peptides and training datasets (L. 349-353, Supplementary Fig. 7b-c, Supplementary Data 15) through NW similarity analysis. We have updated Fig. 4a to clarify the MAI framework in the revised manuscript.

The newly added contents read as follows:

1. L. 706-709: “We applied a global pairwise sequence alignment using the Needleman–Wunsch algorithm with the Bio.pairwise2 module in Python (version 3.9) to estimate the similarity between 11,379 RiPP core peptides and peptides in two training datasets. The sequence identity for each alignment was normalized by the maximum length of the query or subject sequence. For each query, the subject sequence with the highest identity was recorded as the best hit.”

2. L. 349-353: “To confirm that these predictions were made on novel sequences, we performed a pairwise Needleman–Wunsch similarity analysis between the EEMC-derived core peptides and the training datasets. Most RiPP sequences shared low similarity with the global AMP and toxicity training datasets (Supplementary Fig. 7b-c, Supplementary Data 15), indicating that the cAMP predictions were applied to sequences independent of the training datasets.”

5. No assays for serum stability, resistance evolution, or in vivo efficacy are provided, which are standard in the field, further leaving translational claims unsupported by data. I really think testing the peptides in at least an animal infection model is a major issue and critical here.

Response:

Based on your suggestion, we attempted to assess serum stability and resistance evolution of cAMPs to evaluate their translational potential. However, due to the high cost and the stringent institutional requirements for animal experiment ethical approval, *in vivo* efficacy tests in mouse models have not been conducted.

We first attempted serum stability assays. We confirmed that the peptides were able to dissolve in a mixture of water/acetonitrile (ACN) (1:3, v/v), which was used to extract the peptides from the plasma (**Response Fig. 3a**). However, when the peptide cAMP_15 was incubated with plasma and extracted with three proportions of acetonitrile, no peptide signal could be recovered by HPLC at any time point (**Response Fig. 3b**). We then realized that the cationic cAMPs were unable to be extracted from the plasma, which contains many negative charged proteins like albumin. To confirm our hypothesis, we did the recovery assays for two other peptides, cAMP_40 and cAMP_48. The results once again show that no peptide signals were detected by HPLC as long as plasma was added (**Response Fig. 3c-d**). We thought that the acidification of plasma during the extraction step may protonize the carboxylic groups of negatively charged protein and thus help the release of bound cAMPs. However, the addition of 0.1-1% of formic acid or trifluoroacetic acid also does not help. Because the tested cAMPs are strong cationic peptides with a high isoelectric point of nearly 11 (**Supplementary Data 16**), they may bind strongly to negatively charged plasma proteins through electrostatic interactions. As a result, we cannot recover the peptides from the plasma under standard protein-precipitation conditions, and therefore, reliable plasma stability measurements could not be obtained. We note that sequence optimization strategies (e.g., terminal protection, D-amino acid substitution, or cyclization) could mitigate this issue^{9,10}, but such chemical modifications are beyond the scope of the current screening-focused study and will be explored in future work.

We finished longitudinal resistance assay of cAMP_81 against four pathogens, including *E. coli* ATCC 25922, *A. baumannii* ATCC 19606, *A. baumannii* BAA-1605, and *E. faecium* BM4105, using polymyxin B as the positive control. We observed that 0.5×MIC cAMP_81 treatment for 30 days induced less resistance compared with the positive control (**Supplementary Fig. 11d, Supplementary Data 21**). These results further support the potential of cAMP_81 as a membrane-targeting antimicrobial agent.

Response Fig. 3. The plasma stability assays for cAMPs. a, Solubility of cAMPs in water/acetonitrile (1/3, v/v). **b,** Plasma stability of cAMP_15 in 24 hours. **c,** Recovery assay of cAMP_40 from plasma. **d,** Recovery assay of cAMP_48 from plasma.

Supplementary Fig. 11. d, Longitudinal resistance assays for cAMP_81 and polymyxin B against *A. baumannii* ATCC 19606, *E. coli* ATCC 25922, MDR *E. faecium* BM4105, and MDR *A. baumannii* BAA-1605. The y-axis shows the fold change of MIC relative to the initial (day 0) MIC. The x-axis shows the passage points at which MIC was measured (days 0, 10, 20, and 30).

Specific revisions:

We have finished longitudinal resistance assays. The method and corresponding results have been incorporated into the Methods (L. 791–796) and Results section (L. 425–428, L. 429–430), accompanied by new supplementary figures and supplementary Data (Supplementary Fig. 11d, Supplementary Data 21) in the revised manuscript.

The newly added contents read as follows:

1. L. 791–796: “To assess the resistance development during serial passage, *E. coli* ATCC 25922, *A. baumannii* ATCC 19606, *A. baumannii* BAA-1605, and *E. faecium* BM4105 were cultured in MHB at 37 °C overnight. Subsequently, 2 µl of this culture was used to inoculate 200 µl of fresh MHB containing cAMP_81 at a concentration of 0.5×MIC. This process was repeated every 24 h: inoculating 200 µl of fresh MHB containing cAMP_81 at a concentration of 0.5×MIC with the previous culture, at a 1% inoculum ratio. This cycle was continued for 30 days, and MIC assays were performed on the resultant bacteria at 10-day intervals.”

2. L. 425–428: “Notably, longitudinal resistance assays with cAMP_81 against *A. baumannii* ATCC 19606, *E. coli* ATCC 25922, MDR *E. faecium* BM4105, and MDR *A. baumannii* BAA-1605, using polymyxin B as the positive control, demonstrated that 0.5×MIC cAMP_81 treatment for 30 days induced less resistance compared with the control (Supplementary Fig. 11d, Supplementary Data 21).”

3. L. 429–430: “and the low resistance induction observed in the longitudinal assays further supports their potential as membrane-targeting antimicrobial agents.”

6. The authors should provide all accession numbers/DOIs for raw assemblies, gene catalogues, trained model weights, and inference scripts to enable reproducibility of the work.

Response:

Thanks for your important comment. We equally acknowledge the importance of data availability. The raw assemblies (including 74,999 MAGs recovered in this study, along with the 83 in-house isolate genomes from deep-sea), two non-redundant gene catalogues, and 163,693 BGCs can be interactively accessed at the China National GeneBank DataBase (CNCBdb) (<https://db.cngb.org/search/project/CNP0007106/>). The accession IDs of publicly available bacterial and archaeal REFs from NCBI are provided in Supplementary Data 2. The trained model weights are now publicly available at Zenodo (<https://zenodo.org/records/17613552>). The inference scripts to enable reproducibility of the model results are now publicly available on our GitHub

repository (<https://github.com/BGI-METAI/Metagenome-AI>). We have updated the Data availability section (L. 798-802) and Code availability section (L. 809-811) accordingly.

Specific revisions:

We have updated the Data availability (L. 798-802) and Code availability sections (L. 809-811) in the revised manuscript to provide a project accession number for assembled genomes and the gene catalog, as well as trained model weights and inference scripts, ensuring that readers can reproduce the work.

The updated contents read as follows:

1. L. 798-802: “All 74,999 MAGs recovered in this study, together with 83 in-house isolate genomes from deep-sea, the non-redundant gene sets from assembled contigs and genomes, and 163,693 BGCs, can be interactively accessed at the China National GeneBank DataBase (CNCBdb) (<https://db.cngb.org/search/project/CNP0007106/>). The accession IDs of publicly available bacterial and archaeal reference genomes from NCBI are provided in Supplementary Data 2.”

2. L. 809-811: “The trained model weights and corresponding datasets are now publicly available at Zenodo (<https://zenodo.org/records/17613552>). The inference scripts enabling reproduction of the model results are publicly available on our GitHub repository (<https://github.com/BGI-METAI/Metagenome-AI>)”

7. The discussion should benchmark the six potent AMPs against published examples (e.g., prevotellin-2) and compare the MAI framework with prior microbiome-mining pipelines for peptide discovery. Moreover, the authors do not discuss recent mining efforts of archaeal proteomes for bioactive antibiotic peptides.

Response:

We agree that the discussions you suggested may improve the rigor of our manuscript. To benchmark our potent AMPs against published examples, we synthesized prevotellin-2 and synechocucin-1 as positive controls due to their broad antimicrobial activity^{11,12}. We tested their MICs against eight pathogens used in our work. We observed that synechocucin-1 and prevotellin-2 displayed antimicrobial activity against seven and three strains, respectively. Notably, compared with positive controls, cAMP_81 displayed lower or comparable MICs against *A. baumannii* ATCC 19606, *E. coli* ATCC 25922, and MDR *A. baumannii* BAA-1605 (**Supplementary Fig. 11b**).

We further compared the MAI framework with a published and widely used microbiome-mining pipeline, the human cAMP model¹³, which is based on three deep learning algorithms. The MAI framework outperformed the human cAMP model, with the ESM2-3B-based model achieving over 10% improvement in accuracy, F1-score, area under the receiver-operating characteristic curve (AUC), and Matthews correlation coefficient (MCC) (**Supplementary Fig. 8c**). This underscores the utility of large-scale protein language models in decoding complex sequence-function relationships.

We also incorporated a discussion of recent mining efforts of archaeal proteomes using the APEX model¹⁴, highlighting the untapped potential of archaeal microbiomes for antibiotic discovery.

Supplementary Fig. 11. b, Determination of MICs of the seven potent cAMPs and two published AMPs (prevotellin-2 and synechocucin-1) against eight commonly studied pathogens.

Supplementary Fig. 8. c, Classification metrics of different AMP prediction tools, including Macrel, AMPScanner, iAMPpred, amPEPpy, c_AMP model, and our model trained with ESM2-3B, on the global AMP test dataset (5,521 sequences).

Specific revisions:

We have added the benchmarking results (L. 390-394) and updated comparative analyses (L. 334-339) to the Results section, and have added the discussion of recent mining efforts of archaeal proteomes (L. 488-491) in the Discussion section of the revised manuscript.

The newly added and updated contents read as follows:

1. L. 390-394: “In addition, we synthesized two broad-spectrum AMPs (synechocucin-1⁶⁷ and prevotellin-2⁶⁶) as positive controls. We observed that synechocucin-1 and prevotellin-2 displayed antimicrobial activity against seven and three strains, respectively. Notably, compared to positive controls, cAMP_81 displayed lower or comparable MICs against *A. baumannii* ATCC 19606, *E. coli* ATCC 25922, and MDR *A. baumannii* BAA-1605 (Supplementary Fig. 11b).”

2. L. 334-339: “We then compared the performance of pLLM-based models with the popular AMP prediction tools and peptide toxicity model, including Macrel³⁷, AMPScanner³², iAMPpred³³,

amPEP³⁴, cAMP model³⁰, and ToxinPred⁶⁸. The ESM2-3B-based models achieved more than 10% improvement in performance over the known AMP models in terms of accuracy, F1-score, area under the receiver-operating characteristic curve (AUC), and Matthews correlation coefficient (MCC) (Supplementary Fig. 8c, Supplementary Data 14).”

3. **L. 488-491:** “In addition, we identified 6 cAMPs derived from archaeal BGCs, one of which (cAMP_102) exhibited MICs of 64 μ M and 128 μ M against *A. baumannii* ATCC 19606 and MDR *A. baumannii* BAA-1605, respectively. Recent work has revealed archaeal proteomes as a rich source of bioactive peptides³⁹, underscoring the untapped potential of the archaeal microbiomes in antibiotic discovery.”

Minor comments:

1. The ecological or clinical relevance of millions of putative virulence factors is speculative. I suggest highlighting only experimentally validated or high-confidence homologues.

Response:

Thank you for this insightful comment. In the initial analysis, putative virulence factors (VFs) were identified using DIAMOND with a relatively permissive threshold (“blastp e-value 1e-5”), which may indeed capture low-confidence homologues and overestimate VF counts. To address this concern, we re-annotated all VFs using substantially more stringent criteria (“blastp e-value 1e-5 id 40 query-cover 80”). After applying these filters, the total number of predicted VFs decreased from 18,242,431 to 4,340,855, reflecting a major reduction in low-confidence hits and an overall improvement in annotation specificity.

In addition, to avoid speculative interpretations, we have removed the previous discussion regarding potential human health risks, as such conclusions were not sufficiently supported by updated results. Instead, we now contextualize the VF annotations by comparing them with the GEM⁷ and GOMC⁶ datasets to highlight the distinctiveness and novelty of the EEMC gene repertoire, rather than to infer ecological or clinical relevance.

Specific revisions:

We have updated the VF annotations using stringent criteria in the Methods section (**L. 621-626**), removed unsupported interpretations (**L. 212-227**), and added comparative analyses of VFs in the Methods (**L. 626-628**) and Results section (**L. 196-211**) of the revised manuscript.

The updated and added contents read as follows:

1. **L. 621-626:** “We used Prodigal with default parameters for ORF prediction on the MAGs and isolate genomes from extreme environments, as well as the genomes from the GEM and GOMC datasets^{19,47}. The protein sequences of ORFs from the above three datasets were combined and clustered into gene clusters at a threshold of 70% average amino acid identity and 80% AF, using MMseqs2 with the parameters "easy-linclust -e 0.001 -min-seq-id 0.7 -c 0.8", as used in the TG2G¹⁸. The VFs, ARGs, and CAZymes were annotated by aligning the protein sequences of all above ORFs against the VFDB, CARD, and CAZy database, respectively, using DIAMOND with the parameters "blastp e-value 1e-5 id 40 query-cover 80"¹⁰⁰.”

2. **L. 626-628:** “MMseqs2 was used to evaluate the similarity of the VFs from extreme environments with those in the GEM and GOMC datasets. The same approach was applied for

ARGs and CAZymes.”

3. L. 196-211: “To further evaluate the novelty of our gene resources, we predicted 203,220,315 ORFs directly from the 78,213 microbial genomes, and compared them with genome-derived gene datasets from GEM and GOMC. Genes from the three datasets were clustered into 116,034,223 gene clusters, of which 76,824,985 originated from our EEMC gene catalog (Fig. 2c; at thresholds of 70% coverage and 80% AAI). Among these, only 15.68% (12,045,620/76,824,985) clusters were shared with either the GEM or GOMC datasets. Consistent with the novelty distribution of the unigenes, clustering against the GEM and GOMC datasets revealed that the deep-sea environment harbored the largest number of novel gene clusters (27,177,489), whereas the hyperacid environment exhibited the highest proportion of novel clusters (88.16%) (Fig. 2d). We next examined the functional characteristics of genes from the EEMC by performing comparative analyses using the VFDB, CARD, and CAZy databases to identify potential virulence factors (VFs), antibiotic resistance genes (ARGs), and carbohydrate-active enzymes (CAZymes), respectively. For each category, genes identified from the EEMC gene catalog were clustered together with those from the GEM and GOMC gene catalogs using the same criteria (70% coverage and 80% AAI). Specifically, 4,340,855 potential VFs were identified from the EEMC gene catalog, resulting in 1,218,145 VF clusters, of which only 21.48% (261,641/1,218,145) were shared with either the GEM or GOMC datasets (Fig. 2e). Similarly, 269,235 ARG clusters and 3,513,158 CAZyme clusters were obtained from the EEMC gene catalog, with only 19.65% (52,903/269,235) and 17.76% (623,818/3,513,158) of them, respectively, overlapping with the GEM or GOMC datasets (Fig. 2f-g). All these together highlight the distinctiveness of the EEMC gene catalog.”

Fig. 2. c, Venn diagram showing the overlap of genome-derived gene clusters among the EEMC, GEM, and GOMC datasets. The intersections represent shared clusters, while the non-overlapping areas indicate unique clusters specific to each dataset. **d,** Novelty and total number of genome-derived gene clusters identified in each extreme environment. **e-g,** Venn diagrams showing the overlap of genome-derived gene clusters associated with **(e)** VFs, **(f)** ARGs, and **(g)** CAZymes among the EEMC, GEM, and GOMC datasets.

2. Please check the manuscript for typos – e.g. “enviroment” (line 71), “archael” (line 74).

Response:

We have carefully checked the entire manuscript and corrected all typos, including “enviroment” (now “environment”, L. 77) and “archael” (now “archaeal”, L. 80).

Specific revisions:

We have thoroughly proofread the revised manuscript and corrected the specific typographical errors mentioned by the reviewer, as well as additional minor spelling and grammar inconsistencies identified throughout the text.

3. The authors mention the lack of generalizable models, but this is not entirely accurate (e.g., APEX).

Response:

We agree that several recent deep learning models, such as the human cAMP model¹³ and APEX^{14,15}, indeed demonstrate better generalizability compared to earlier AMP predictors. We have revised the Introduction section to accurately reflect this point by acknowledging these advances while clarifying that such models are still trained on relatively limited AMP and non-AMP peptide datasets and do not leverage protein large language models. This distinction highlights the motivation for developing a pLLM-based framework in our study.

Specific revisions:

We have revised the Introduction section to (i) acknowledge existing generalizable models, including APEX and the human cAMP model, and (ii) clarify their reliance on curated peptide datasets rather than pLLM-based representations (L. 67-70).

The newly added content reads as follows:

L. 67-70: “Although several recent deep learning models, such as the human cAMP model³⁰ and APEX^{38,39}, demonstrate improved generalization, they are still trained primarily on curated AMP and non-AMP peptide datasets rather than on large-scale protein representations, and therefore do not leverage the broad sequence–function priors captured by protein language models.”

4. References. Check for issues. E.g., update citation 37 (Macrel) year (should be 2020, not 2010).

Response:

We have corrected the publication year of citation 37 (Macrel) from 2010 to 2020 (L. 879-880). We have also carefully reviewed all references in the manuscript to ensure accuracy and consistency.

Specific revisions:

We have updated citation 37 with the correct publication year (2020) and verified other references for accuracy and formatting issues.

Reviewer #2 (Remarks to the Author):

The manuscript by Jiang and co-authors contributes a significant resource to the scientific community with the generation of the Extreme Environment Microbiome Catalog (EEMC) as well as making an important contribution to the field of antimicrobial peptides and genome mining for these bioactive compounds. The authors collected publicly available (meta)genomes from extreme environments, assembled MAGs from these, then analyzed them (including some pure isolate genomes from their own collection) for the presence of BGCs, with an emphasis on RiPPs. Separately they created a curated database of antimicrobial peptides (active against either Gram + or Gram - bacteria or with broad spectrum), but non-toxic, pulled from available databases. This was used as training and validation data for their custom machine learning model (Metagenome-AI) used to develop classifiers capable of detecting peptides with these desirable parameters from a pool of uncharacterized RiPPs, mined from the MAGs they reconstructed. Next, they selected 100 of the peptides predicted to be antimicrobial to be chemically synthesized. These peptides were assayed for their antimicrobial activity against a range of Gram + and Gram - bacteria. These experiments revealed that a high proportion of the peptides selected (84/100) demonstrated antimicrobial activity and seven of these were demonstrated with a CC50 higher than their MIC suggesting that they have therapeutic potential.

Response:

We are deeply grateful for your constructive feedback and the thorough and accurate summary of our study. We believe that the EEMC will be instrumental in advancing genetic resources investigation. We have made several revisions based on your thoughtful comments and suggestions, which have assisted us in improving our manuscript.

General Comments / Corrections to be addressed:

1. In lines 162–164, the authors suggest that microorganisms in extreme environments may gain a competitive advantage due to the widespread presence of secondary metabolite pathways. While this is a well-established ecological principle across diverse habitats, the current analysis does not include comparative data from non-extreme environments, making it difficult to determine whether this trait is uniquely enriched or more pronounced in the sampled extremes. The statement may unintentionally imply such uniqueness, and a brief clarification would help ensure the intended nuance is preserved. Alternatively, provide references to support the idea that this is particularly enriched in extreme environments, if that is what the authors intended.

Response:

We acknowledge that our previous analysis did not include a formal comparison with non-extreme environments. Our intention was not to suggest that microorganisms in all extreme environments possess secondary metabolite pathways to a uniquely enriched degree. Instead, our interpretation was based strictly on patterns observed within the EEMC dataset, specifically the large number of uncharacterized genes and the abundant representation of secondary metabolism-related functions. These patterns highlight the potential of the studied extreme habitats, as well as

the EEMC itself, as sources of novel natural products. To avoid unintended implications, we have revised the text in the Result section (L. 181-188) to clarify this point and removed language that could be interpreted as suggesting comparative enrichment.

Specific revisions:

We have revised the text in the Result section (L. 181-188) to clarify the widespread presence of secondary metabolite pathways and removed language that could be interpreted as suggesting comparative enrichment.

L. 181-188: The original context “Moreover, we observed the widespread presence of pathways related to the biosynthesis of antibiotics and secondary metabolites (ko01130, ko01110) (Supplementary Fig. 3d and Supplementary Data 4), suggesting that microorganisms may gain competitive advantage in extreme environments by inhibiting competitors and modulating interspecies interactions, making the microbes in these habitats potentially rich reservoirs of novel natural products.” was revised to “Moreover, we observed the widespread presence of pathways related to the biosynthesis of antibiotics and secondary metabolites (ko01130, ko01110) (Supplementary Fig. 3d and Supplementary Data 4), suggesting that microorganisms from extreme environments likely employ chemical strategies for competition and interaction, an ecological mechanism broadly observed across diverse microbial ecosystems^{2,11,61}. Given the considerable number of novel genes and biosynthetic features uncovered in the EEMC, the microbes in these habitats may therefore represent valuable reservoirs of novel natural products.”

2. The paragraph spanning lines 174–189 places considerable emphasis on the potential threat to human health posed by virulence factors. However, this framing seems at odds with several findings that highlight the novelty and niche specificity of the microbial genomes and functions: (i) the remarkably limited sharing of unigenes across environments; (ii) low percentage similarity to the GEM and UHGG databases; (iii) biosynthetic gene cluster (BGC) analysis indicating a high proportion of habitat-specific GCFs and GCCs; and (iv) the identification of highly novel virulence factors. These results support the authors’ conclusion that strong environmental selective pressures drive the evolution of microbial lineages toward peptides with specialized ecological roles. In this context, virulence appears to be highly adapted to specific environmental conditions and may not be directly translatable to human pathogenicity. A more balanced framing would better reflect the ecological significance of these findings. However, the rationale for prioritising this analysis over other metabolic traits is not clear. As presented, it does not substantially advance the study’s main focus. To strengthen the manuscript’s coherence, the authors might consider either omitting this section or, alternatively, expanding the scope to include comparable analyses of other functional traits. This would provide a more balanced and contextually grounded assessment of microbial adaptation mechanisms in extreme environments.

Response:

Thanks for your constructive and insightful evaluation. We agree that our previous framing of virulence factors (VFs) overemphasized potential human health risks, whereas our analyses were not aiming to provide such a conclusion. Our results, including limited sharing of unigenes across extreme environments, the low sequence similarity to known gene catalogs (GEM and UHGG), and

the remarkable novelty of the predicted VFs, collectively indicate that the VFs identified in the EEMC are shaped primarily by environment-driven functional diversification, rather than reflecting human-pathogenic potential. Accordingly, we have removed statements implying human health in the revised manuscript (L. 212-227).

To provide a more balanced assessment of the EEMC gene catalog, we have expanded our functional analyses to include antibiotic resistance genes (ARGs, annotated via CARD) and carbohydrate-active enzymes (CAZymes, annotated via CAZy), in addition to VFs. Comparative clustering across EEMC, GEM⁷, and GOMC⁶ datasets shows that a substantial fraction of functional gene clusters in the EEMC are novel (L. 626-628, L. 196-211).

Specific revisions:

We have removed statements implying human health implications (L. 212-227) and added comparative analyses of VFs in the Methods (L. 626-628) and Results section (L. 196-211) of the revised manuscript.

The added contents read as follows:

1. L. 626-628: “MMseqs2 was used to evaluate the similarity of the VFs from extreme environments with those in the GEM and GOMC datasets. The same approach was applied for ARGs and CAZymes.”

2. L. 196-211: “To further evaluate the novelty of our gene resources, we predicted 203,220,315 ORFs directly from the 78,213 microbial genomes, and compared them with genome-derived gene datasets from GEM and GOMC. Genes from the three datasets were clustered into 116,034,223 gene clusters, of which 76,824,985 originated from our EEMC gene catalog (Fig. 2c; at thresholds of 70% coverage and 80% AAI). Among these, only 15.68% (12,045,620/76,824,985) clusters were shared with either the GEM or GOMC datasets. Consistent with the novelty distribution of the unigenes, clustering against the GEM and GOMC datasets revealed that the deep-sea environment harbored the largest number of novel gene clusters (27,177,489), whereas the hyperacid environment exhibited the highest proportion of novel clusters (88.16%) (Fig. 2d). We next examined the functional characteristics of genes from the EEMC by performing comparative analyses using the VFDB, CARD, and CAZy databases to identify potential virulence factors (VFs), antibiotic resistance genes (ARGs), and carbohydrate-active enzymes (CAZymes), respectively. For each category, genes identified from the EEMC gene catalog were clustered together with those from the GEM and GOMC gene catalogs using the same criteria (70% coverage and 80% AAI). Specifically, 4,340,855 potential VFs were identified from the EEMC gene catalog, resulting in 1,218,145 VF clusters, of which only 21.48% (261,641/1,218,145) were shared with either the GEM or GOMC datasets (Fig. 2e). Similarly, 269,235 ARG clusters and 3,513,158 CAZyme clusters were obtained from the EEMC gene catalog, with only 19.65% (52,903/269,235) and 17.76% (623,818/3,513,158) of them, respectively, overlapping with the GEM or GOMC datasets (Fig. 2f-g). All these together highlight the distinctiveness of the EEMC gene catalog.”

Fig. 2. c, Venn diagram showing the overlap of genome-derived gene clusters among the EEMC, GEM, and GOMC datasets. The intersections represent shared clusters, while the non-overlapping areas indicate unique clusters specific to each dataset. **d,** Novelty and total number of genome-derived gene clusters identified in each extreme environment. **e-g,** Venn diagrams showing the overlap of genome-derived gene clusters associated with **(e)** VFs, **(f)** ARGs, and **(g)** CAZymes among the EEMC, GEM, and GOMC datasets.

3. In line 182, the authors report up to 1,191 virulence factors within a single genome. This figure appears unusually high and may warrant further context. Could the authors comment on whether such a number is typical, or whether it might reflect an overestimation due to annotation thresholds or database limitations? It would be helpful to reference comparable cases in the literature where similarly high counts of virulence factors have been reported, to support the plausibility of this finding.

Response:

We appreciate your observation regarding the unusually high number of VF identified within individual genomes. In our previous analysis, VF annotation was performed using VFDB (version 2024) and DIAMOND with the permissive threshold “blastp e-value 1e-5”, following the parameters adopted in the TG2G and TPMC^{2,16}. Under these settings, the highest VF count observed in the EEMC was 1,191 within a single genome. For comparison, TG2G, which used VFDB (version 2019) and identical mapping criteria, reported up to 374 VFs in a genome. Given the substantial expansion of VFDB in recent years, the higher value we observed was within a conceptually plausible range.

To further assess the plausibility of such high counts, we re-annotated VFs in several public genome catalogs, including TG2G, GEM, and GOMC, using the same database and parameters as applied in the EEMC. We observed up to 1,061, 1,024, and 1,270 VFs in individual genomes of TG2G, GEM, and GOMC, respectively (**Response Table 2**). These results indicate that the high VF counts are not unique to our dataset but likely reflect database expansion and permissive alignment settings.

In response to this concern, and consistent with the minor comment #1 from Reviewer #1, we applied a more stringent alignment threshold (“blastp e-value 1e-5 id 40 query-cover 80”). This adjustment substantially reduced the total number of predicted VFs from 18,242,431 to 4,340,855 across EEMC genomes. For the genome initially annotated with 1,191 VFs, the count decreased to 168 under the stricter threshold. Across the entire dataset, the genome with the highest VF count carried 521 VFs, whereas the same genome carried 706 VFs under the previous threshold (**Response Table 2, Response Fig. 4**). Consistent trends were observed in TG2G, GEM, and GOMC, where the highest VF counts were 487, 382, and 479 VFs, respectively (**Response Table 2, Response Fig. 4**).

Together, these results suggest that the originally high VF counts are primarily driven by permissive annotation thresholds rather than true biological enrichment. The revised, conservative parameters now provide a more accurate and reliable estimation of VF counts across genomes in the EEMC. Corresponding updates have been incorporated into the Methods and Results sections of the revised manuscript.

Catalog	Genome counts	Max VF counts (blastp e-value 1e-5)	Max VF counts (blastp e-value 1e-5 id 40 query-cover 80)
EEMC	78,213	1,191	521
TG2G	3,241	1,061	487
GEM	22,732	1,024	382
GOMC	43,191	1,270	479

Response Table 2. The max VF counts using different alignment thresholds in four genome catalogs.

Response Fig. 4. The VF counts identified in genome four genome catalogs using different alignment thresholds.

Specific revisions:

We have re-annotated VFs in all EEMC genomes using stricter criteria (“blastp e-value 1e-5 id 40 query-cover 80”), which substantially reduced inflated VF counts and improved annotation specificity. We have also evaluated VF counts in public genome catalogs (TG2G, GEM, and GOMC) under identical parameters and VFDB version to confirm that high VF counts are not unique to the EEMC. We have revised the alignment criteria (L. 624-626) and results (L. 203-208) in the Methods and Results sections of the revised manuscript.

The revised contents read as follows:

1. L. 624-626: “The VFs, ARGs and CAZymes were annotated by aligning the protein sequences of all above ORFs against the VFDB, CARD and CAZy database, respectively, using DIAMOND with the parameters “blastp e-value 1e-5 id 40 query-cover 80”¹⁰⁰.”

2. L. 203-208: “We next examined the functional characteristics of genes from the EEMC by performing comparative analyses using the VFDB, CARD, and CAZy databases to identify potential virulence factors (VFs), antibiotic resistance genes (ARGs), and carbohydrate-active enzymes (CAZymes), respectively. For each category, genes identified from the EEMC gene catalog were clustered together with those from the GEM and GOMC gene catalogs using the same criteria (70% coverage and 80% AAI). Specifically, 4,340,855 potential VFs were identified from the EEMC gene catalog, resulting in 1,218,145 VF clusters, of which only 21.48% (261,641/1,218,145) were shared with either the GEM or GOMC datasets (Fig. 2e).”

4. The authors provided access to the MAGs, as well as the software pipeline they created, but not the final curated list of peptides used as training the dataset nor the final trained model. If these are not to be patented, it would be a fantastic resource for the rest of the microbial and peptide community in particular. Could the authors comment on why these were not made available?

Response:

We appreciate your emphasis on reproducibility and community reuse. At the time of the initial submission, we did not release the curated peptide list and trained classifiers because these resources were undergoing internal evaluation for potential intellectual property considerations. This assessment has since been completed, and no patent application was pursued. Accordingly, we have now made both the curated peptide dataset and the frozen trained models publicly available under permissive academic licenses. These resources are versioned, permanently archived, and citable on Zenodo (<https://zenodo.org/records/17613552>).

Specific revisions:

We have updated the Code availability section (L. 809-811) to include information regarding the access to curated peptide datasets and trained models. We hope this update addresses the reviewer’s concern and enables immediate reuse of the datasets and trained models.

The updated content reads as follows:

L. 809-811: “The trained model weights and corresponding datasets are now publicly available at Zenodo (<https://zenodo.org/records/17613552>). The inference scripts enabling reproduction of the model results are publicly available on our GitHub repository (<https://github.com/BGI-METAI/Metagenome-AI>)”

5. What is the distribution of the 108 cAMPs between MAGs and isolates?

Response:

Among the 108 cAMPs, 88 were predicted from MAGs and 20 from isolate genomes. The full information for all 108 cAMPs has now been provided in **Supplementary Data 15**, where we have added a dedicated column (“synthesis”) indicating which peptides were successfully synthesized and which were not.

Specific revisions:

We have added a description of the distribution of the 108 cAMPs (MAG-derived vs. isolate-derived) in the Results section (**L. 355-358**). In addition, we have included a new supplementary table (**Supplementary Data 15**) that provides detailed information for all 108 cAMPs.

The newly added content reads as follows:

L. 355-358: “Of the 108 cAMPs, 88 cAMPs were predicted from MAGs, and 20 were predicted from isolates (Supplementary Data 15). All 108 selected linear cAMPs were shorter than 30 amino acids and lacked post-translational modifications (PTMs), due to the limited reliability of PTM prediction^{29,69} and length constraints in chemical synthesis.”

6. While acknowledging the cost and logistical constraints, it would have strengthened the study if the authors had also synthesized and assayed a comparable number of peptides from the EEMC predicted to be non-antimicrobial. This would have provided a more balanced assessment of the predictive model’s specificity, demonstrating its ability not only to identify true positives but also to exclude false positives.

Response:

We agree that evaluating model-predicted negatives provides an orthogonal assessment of specificity. However, due to financial constraints, synthesizing a negative set comparable in size to the 100 cAMPs was not feasible at this stage.

Nonetheless, to address this concern within the available resources, we synthesized 20 peptides that our classifier predicted to be non-antimicrobial and tested them under the same experimental conditions used for the cAMPs. Among these 20 predicted non-AMPs, only three peptides showed inhibition against at least one pathogen (relative $OD_{600} < 0.8$, **Supplementary Fig. 11a**, **Supplementary Data 17**), corresponding to a false negative rate of 15%. Although this smaller negative set does not allow a full evaluation of specificity, it provides an empirical estimate of the model’s ability to avoid incorrectly excluding active peptides. In parallel, our high positive predictive rate within the synthesized cAMP set (84 out of 100 peptides showing activity) demonstrates strong enrichment for true positives. Together, these complementary assessments provide meaningful experimental support for model robustness despite the limited scale of negative peptide synthesis. We have added these results in the revised Results (**L. 375-378**) and Methods sections (**L. 711-712**).

Supplementary Fig. 11. a, The heatmap shows the relative OD₆₀₀ values of the cells grown in medium supplemented with candidate non-AMPs at a concentration of 60 μ M, normalized to the OD₆₀₀ of the cells grown in unsupplemented medium. Only the relative OD₆₀₀ below 80% is labeled.

Specific revisions:

We have synthesized 20 candidate non-AMPs that our classifier predicted to be non-antimicrobial and tested their antimicrobial activity under the same antimicrobial assays used for the cAMP. We have updated the method and results in the Methods (L. 711-712) and Results section (L. 375-378), respectively.

The updated contents read as follows:

1. L. 711-712: “All the cAMPs, non-AMPs, and two published AMPs (synechocucin-1 and prevotellin-2) used in this study were synthesized via solid-phase peptide synthesis by Sangon Biotech.”

2. L. 375-378: “Simultaneously, a negative set of twenty peptides was chosen from a predicted non-AMP set, of which three were effective in inhibiting at least one strain, indicating that our method has a low false negative rate of 15% (Supplementary Fig. 11a, Supplementary Data 17). Together, these assessments provided meaningful experimental support for model robustness despite the limited scale of peptide synthesis.”

7. Many RiPPs derive their bioactivity from post-translational modifications (PTMs) such as glycosylation and stereoselective modifications. Given that 96 of the 100 synthesized cAMPs were classified as PTM peptides (lassopeptides and class II lanthipeptides), did the synthetic synthesis incorporate these PTMs and what structural validations were performed beyond mass confirmation and purity? This could be a critical gap, as PTMs, especially those conferring conformational rigidity like disulfide bridges or macrocyclization, are essential for the biological activity of RiPPs. Given that DeepRiPP was used to predict core peptides independently of genomic context, and that most synthesized peptides were lassopeptides and lanthipeptides, it is likely that the predicted sequences represent only the core scaffold, not the full PTM landscape. Therefore, the authors should clarify whether synthetic strategies

attempted to mimic native PTMs (e.g., via chemical cyclization or enzymatic modification), and whether stereochemistry was verified (e.g., via NMR or circular dichroism). Without such validation, the observed cytotoxicity and antimicrobial activity may not fully reflect the native peptide's properties, potentially limiting translational relevance. This does not detract from the impressive success rate; however, it may help explain why the observed activities predominantly reflect similar modes of action and lack novelty in terms of distinct scaffolds or mechanistic diversity.

Response:

Thank you for raising this point. We would like to clarify that PTMs were not explicitly modeled or encoded during our machine learning (ML) training. The pLLM-based model was trained solely on amino acid sequences without structural or modification annotations, and the core peptides predicted by DeepRiPP correspond to the genetically encoded precursor regions prior to any post-translational processing¹⁷. Therefore, the outputs of models represent unmodified peptides rather than mature, post-translationally modified RiPPs. In this first breadth-oriented screen, we synthesized the unmodified core peptides predicted from RiPP BGCs; no PTMs (e.g., macrocyclization and dehydrations/lanthionine linkages¹⁸) were installed. The decision was driven by (i) the current uncertainty in inferring exact PTM patterns/sites from precursor sequences alone^{17,19} and (ii) the cost for chemoenzymatic or total-chemical installation of PTMs at the scale of this study (96/100 were lassopeptide or class II lanthipeptide core peptides). We agree that PTMs and stereochemical control are often essential for investigating the native bioactivity of RiPPs.

We have clarified these limitations in the revised manuscript (L. 506-519) and tempered claims accordingly: our aim here was to demonstrate that pLLM-guided mining of core peptides can prospectively enrich positive candidates for antimicrobial activity at scale, not to reconstitute the full PTM landscape or establish scaffold-level mechanism. We also have acknowledged in the Discussion section that the absence of PTMs likely narrowed chemical-space exploration and may help explain the observed convergence in modes of action. Going forward, we will try to predict enzymatic PTM sites from isolate/MAG BGCs using deep learning models, install the corresponding PTMs, and verify stereochemistry, enabling head-to-head activity comparisons with unmodified cores.

To provide structural context without over-claiming, we first used AlphaFold3 for all synthesized cAMPs and non-AMPs, and summarized secondary-structure propensities with DSSP. Among 100 cAMPs, 89 showed α -helical propensity, while 7 non-AMPs showed α -helical propensity (Fig. 6a, Supplementary Data 16). These analyses are presented exploratorily and do not substitute for state-specific structural validation. We then added circular dichroism (CD) measurements for 7 active peptides (those with defined MICs) in water and trifluoroethanol (TFE) mixtures (3:2, v/v), providing experimental secondary-structure readouts. Among 7 selected cAMPs, 5 cAMPs showed α -helical propensity (Fig. 6b, Supplementary Fig. 11c). This structural tendency may underline the membrane-interacting mechanisms commonly observed in α -helical antimicrobial peptides^{13,20}. We have added the method and results (L. 760-766, L. 406-411, L. 411-415) in the revised manuscript.

Fig. 6. a, AlphaFold3 was used to generate structural predictions using default parameters. Three-dimensional structures of the resulting PDB files were generated using Mol*3D Viewer. **b,** CD results for the seven cAMPs with MIC values.

Supplementary Fig. 11. c, The CDNN results of the CD assays for the 7 selected cAMPs.

Specific revisions:

We have added structure predictions (L. 406-411) of cAMPs and non-AMPs, and CD measurements for 7 cAMPs (L. 411-415, L. 760-766) in the revised manuscript. We have clarified the limitations of unmodified RiPPs (L. 506-519) in the revised manuscript.

The updated contents read as follows:

1. L. 406-411: “The secondary structures of peptides are associated with their antimicrobial and other biological activities. To provide structural context without over-claiming, we first predicted the tertiary structures of the 100 synthesized cAMPs and 20 non-AMPs using AlphaFold3⁷¹, and summarized their secondary-structure propensities using the Define Secondary Structure of Proteins (DSSP) algorithm⁷². Among the 100 cAMPs, 89% showed α -helical structures, higher than the proportion (7/20, 35%) in 20 non-AMPs (Supplementary Data 16). The predicted structures of 7 cAMPs with MIC values were shown in the Fig. 6a.”

2. L. 411-415: “We further probed the secondary structure tendencies of 7 cAMPs with MIC

values through circular dichroism (CD) in water and trifluoroethanol (TFE) mixtures (3:2, v/v). We observed 5 helix-dominated peptides (Fig. 6b). CDNN analysis showed that cAMP_15 had the highest α -helical fraction (83.3%), while the remaining helix-containing peptides showed almost 30%, and non-helical peptides showed only 17% (Supplementary Fig. 11c). This structural tendency may underlie the membrane-interacting mechanisms commonly observed in α -helical antimicrobial peptides^{30,40}.”

3. L. 760-766: “CD experiments were performed using a Chirascan V100 (Applied Photophysics, England) at the Research Center for Deepsea Bioresources, Sanya. The experiments were carried out at a temperature of 25 °C in triplicate. Spectra collections were obtained using a quartz cuvette with an optical path length of 0.5 mm, covering a wavelength range from 260 to 190 nm at a rate of 50 nm/min and a bandwidth of 1 nm. The peptides were tested at a concentration of 125 μ M. Measurements were performed in a mixture of water and TFE at a ratio of 3:2 (v/v). Baseline measurements were recorded prior to each measurement. To minimize background effects, a Fourier transform filter was applied. The helical fraction and other fraction values were calculated using the CDNN (version 2.1)⁷².”

4. L. 506-519: “We acknowledge that the peptides synthesized in this study may differ significantly from their native counterparts in folding, stability, and bioactivity. The pLLM-based model was trained exclusively on amino acid sequences without structural or modification annotations, and the core peptides predicted by DeepRiPP represent the genetically encoded precursor regions prior to enzymatic processing. Consequently, the outputs of the MAI framework correspond to unmodified peptide scaffolds rather than mature RiPPs. This decision reflects both the current uncertainty in accurately inferring PTM patterns and sites from precursor sequences alone^{27,29,69} and the prohibitive cost for chemoenzymatic or total-chemical installation of PTMs at the scale of this study (96 of the 100 synthesized cAMPs were lassopeptide or class II lanthipeptide core peptides). Although PTMs and stereochemical constraints are often essential to the native bioactivity of RiPPs, the goal of the present study was to demonstrate that pLLM-guided mining of core peptides can prospectively enrich for antimicrobial activity at scale, rather than to reconstruct native PTM architectures or define mechanisms at the scaffold level. The absence of PTMs likely restricts the accessible chemical space of the synthesized peptides and may partly explain the observed convergence in modes of action of cAMPs. Going forward, as curated PTM datasets become more standardized and accessible, incorporating such information with deep learning methods would enable the prediction of enzymatic PTM sites, validation of stereochemistry, and head-to-head activity comparisons with unmodified cores.”

8. Building on the previous point, the current AMP prediction framework relies predominantly on sequence embeddings without incorporating structural or physicochemical modeling. This approach may overlook context-dependent activity and risks misclassifying peptides with atypical conformations or noncanonical features. Could the authors expand on how the PTMs were accommodated in their ML training? Furthermore, the authors should acknowledge that the predicted peptides may differ significantly from their naturally occurring counterparts in terms of folding, stability, and bioactivity. To clarify the scope and applicability of the models, it is important to specify whether the training datasets were derived from synthetically produced or naturally sourced peptides, and whether the associated activity data reflect in vitro assays or native biological contexts. Furthermore, the

authors are encouraged to discuss how structure-based modeling and molecular dynamics simulations could be integrated to complement sequence-based predictions, thereby enhancing the reliability and interpretability of AMP discovery.

Response:

Thank you for this insightful and constructive comment. We agree that our MAI framework relies primarily on sequence embeddings and does not explicitly incorporate structural or physicochemical modeling. The pretrained protein language models (pLLMs) used in this study (e.g., ESM and ProteinTrans) were trained on millions of natural protein sequences using masked language modeling, enabling them to implicitly capture secondary and tertiary structure as well as residue coevolution²¹⁻²³.

In this study, we constructed antimicrobial peptide (AMP) and toxicity prediction models separately. The AMP dataset was compiled from eight public databases, including APD, CAMP, DBAASP, DRAMP, dbAMP, LAMP, and BaAMP. These databases include both synthetic and naturally derived peptides. Most reported activity measurements derive from *in vitro* assays, with a minority from *in vivo* studies. The toxicity dataset was collected from Hemolytik, DBAASP, and ToxinPred 3.0. However, metadata on peptide origin and experimental context are incomplete for many sequences. To maintain sufficient data diversity, we did not stratify peptides by source or experimental context. We have clarified these points (L. 655-657, L. 675-676) in the revised manuscript.

As noted in our response to your #Comment 7, the core peptides predicted by DeepRiPP and used in our study correspond to unmodified, linear precursor sequences, rather than mature, post-translationally modified RiPPs. In the MAI framework, all sequences were represented in linear, unmodified form without explicit PTM annotations. The currently available AMP and toxicity datasets provide only limited or inconsistent PTM annotations, typically without standardized information on modification sites or types, and therefore, this information was not encoded in the models. Representing all sequences in a uniform linear form ensured consistency and avoided biases from incomplete modification data.

We acknowledge that the MAI framework could be further extended in several ways. We have added a discussion in the Discussion section of the revised manuscript (L. 496-505) on how sequence-based predictions can be complemented by structure-informed features. In future work, we plan to integrate structure-informed representations (e.g., AlphaFold3²⁴-derived embeddings) to complement sequence embeddings, and supplement molecular dynamics simulations to evaluate peptide-membrane interactions, thereby improving model interpretability and predictive reliability. In addition, the MAI framework could be suited for the development and iterative optimization of AMPs with more specific characteristics, such as antimicrobial selectivity and specificity, serum stability, and *in vivo* bioactivity.

Specific revisions:

We have clarified the source and experimental context of collected peptides in the revised manuscript (L. 655-657, L. 675-676). We have added a discussion on how to extend the MAI framework in the revised manuscript (L. 496-505).

The updated contents read as follows:

1. L. 655-657: “These databases included both synthetic and naturally derived peptides, and

the antimicrobial activity data included both *in vitro* assays and *in vivo* experiments. To ensure adequate sample size, we did not stratify peptides by origin or experimental context.”

2. **L. 675-676:** “Most peptides lacked explicit metadata regarding their biological origin or experimental context; therefore, to maintain sufficient data diversity, we did not stratify peptides by source or experimental context.”

3. **L. 496-505:** “While the MAI framework represents a major advancement over previous approaches, it can be further extended in several ways. First, the MAI framework relies on sequence embeddings without explicit incorporation of structural or physicochemical modeling. The pLLMs we employed, including ESM series and ProteinTrans, were black-box models by lacking interpretability, which hinders feature characterization of bioactive peptides. Integrating three-dimensional structure-informed representations derived from popular models, such as AlphaFold3⁷¹, may further improve the performance of the MAI framework and model interpretability. Additionally, to achieve more high-throughput filtering of peptides and improve reliability, the stability of predicted structures and peptide-membrane interactions remain to be validated based on physically based methods, such as molecular dynamics (MD) simulations^{35,36}. The MAI framework is also well-suited for the development and iterative optimization of AMPs with more specific characteristics, such as antimicrobial selectivity and specificity, serum stability, and *in vivo* bioactivity.”

Minor Corrections

1. Ln67: delete “s” from sequences, it should be “sequence datasets”

Response:

We thank the reviewer for pointing this out. The text has been revised accordingly (**L. 73**).

Specific revisions:

We have corrected the typo “sequences datasets” to “sequence datasets” (**L. 73**).

2. Ln147: add a space between “plateauing” and (Supplementary....)

Response:

We thank the reviewer for raising this point. We have added a space between “plateauing” and “(Supplementary ...)” (**L. 165**).

Specific revisions:

We have added a space between “plateauing” and “(Supplementary ...)” in the revised manuscript (**L. 165**).

3. Ln160: metabolic regulation

Response:

We thank the reviewer for pointing this out. The text has been revised accordingly (**L. 179**).

Specific revisions:

We have corrected the typo “metabolic regulate” to “metabolic regulation” (L. 179).

4. Ln239: niche-specificity (not niche-specific)

Response:

We thank the reviewer for raising this point. As part of our revision addressing Reviewer #1 Comment #2, we removed the entire sentence containing this term. Therefore, this specific correction is no longer applicable (L. 190-191).

Specific revisions:

We have removed the sentence containing “niche-specific” in the revised manuscript (L. 190-191).

5. Line 265: Linear azole-containing peptides

Response:

We thank the reviewer for raising this point. The text has been revised accordingly (L. 319).

Specific revisions:

We have corrected the typo “azoele-containing” to “azole-containing” (L. 319).

6. Supplementary figure6 – figure legend, two commas after e-f and are the figures in e and f duplicates (f does not refer to genera and the curves are identical)?

Response:

We thank the reviewer for carefully pointing out this issue. The duplicated panels in the previous version of **Supplementary Fig. 6e–f** resulted from an inadvertent copy–paste error during figure assembly. During the revision, one supplementary figure was removed, and an additional panel was added to this figure, which led to changes in panel numbering (now **Supplementary Fig. 5f–g**). We have replaced the duplicated panel with the correct plot and confirmed that each panel now corresponds to the intended description.

Supplementary Fig. 5. f-g, Rarefaction curves of the top 20 phyla (f) and top 20 genera (g) with the largest numbers of predicted GCFs.

Specific revisions:

We have replaced the duplicated panel with the correct figure and updated the figure legend accordingly (**Supplementary Fig. 5f-g**).

Reviewer #2 (Remarks on code availability):

Yes, I could install and run the code and the authors provide a small dataset to test the program with. Because the authors don't make their database of peptides available, it was not possible to replicate the results in the manuscript. We would strongly suggest that the authors consider making the peptide database available, as well as the trained model which could be a valuable resource for the community.

Response:

As noted in our response above, we have now made all relevant resources publicly available on Zenodo (<https://zenodo.org/records/17613552>), including the peptide data used for training, validation, and testing, the trained models, as well as 11,379 core peptides for prediction. Providing these data ensures full reproducibility of our results and offers a community resource for further method development and benchmarking in AMP discovery. We hope these materials will facilitate broader use and extension of our MAI framework.

Specific revisions:

We have updated the Code availability section (**L. 809-811**) to include the Zenodo repository link (<https://zenodo.org/records/17613552>), which now provides all training, validation, and test peptide datasets, the trained models, and the full set of 11,379 core peptides used for prediction.

The updated content reads as follows:

L. 809-811: “The trained model weights and corresponding datasets are now publicly available at Zenodo (<https://zenodo.org/records/17613552>). The inference scripts enabling reproduction of the model results are publicly available on our GitHub repository (<https://github.com/BGI-METAI/Metagenome-AI>)”

Reviewer #3 (Remarks to the Author):

Response:

We thank you and the co-reviewer for your time and contribution to the evaluation and improvement of our manuscript, and we appreciate the *Nature Communications* initiative to support training in peer review.

Reviewer #3 (Remarks on code availability):

Yes, I could install and run the code and the authors provide a small dataset to test the program with. Because the authors don't make their database of peptides available, it was not possible to replicate the results in the manuscript. We would strongly suggest that the authors consider making the peptide database available, as well as the trained model which could be a valuable resource for the community.

Response:

We thank you for your helpful comment. In response, we have now made all peptide datasets (training, validation, and test sets), the trained models, and the full set of 11,379 core peptides available through a Zenodo repository (<https://zenodo.org/records/17613552>). The updated Code availability section (L. 809-811) now includes the permanent link to this resource. We hope that making these materials publicly accessible will facilitate reproducibility and benefit the community.

Specific revisions:

We have updated the Code availability section (L. 809-811) to include the Zenodo repository link containing all peptide datasets (training, validation, and test), the trained models, and the full set of 11,379 core peptides used for prediction.

The updated content reads as follows:

L. 809-811: “The trained model weights and corresponding datasets are now publicly available at Zenodo (<https://zenodo.org/records/17613552>). The inference scripts enabling reproduction of the model results are publicly available on our GitHub repository (<https://github.com/BGI-METAI/Metagenome-AI>)”

References

1. Bowers, R. M. *et al.* Minimum information about a single amplified genome (MISAG) and a metagenome-assembled genome (MIMAG) of bacteria and archaea. *Nature Biotechnology* **35**, 725 (2017).
2. Cheng, M. *et al.* A genome and gene catalog of the aquatic microbiomes of the tibetan plateau. *Nature Communications* **15**, 1438 (2024).
3. Ma, B. *et al.* A genomic catalogue of soil microbiomes boosts mining of biodiversity and genetic resources. *Nature Communications* **14**, 7318 (2023).
4. Paoli, L. *et al.* Biosynthetic potential of the global ocean microbiome. *Nature* **607**, 111–118 (2022).
5. Delmont, T. O. *et al.* Nitrogen-fixing populations of Planctomycetes and Proteobacteria are abundant in surface ocean metagenomes. *Nature Microbiology* **3**, 804–813 (2018).
6. Chen, J. *et al.* Global marine microbial diversity and its potential in bioprospecting. *Nature* **633**, 371–379 (2024).
7. Nayfach, S. *et al.* A genomic catalog of Earth’s microbiomes. *Nature Biotechnology* **39**, 499–509 (2021).
8. Parks, D. H. *et al.* GTDB: an ongoing census of bacterial and archaeal diversity through a phylogenetically consistent, rank normalized and complete genome-based taxonomy. *Nucleic Acids Research* **50**, D785–D794 (2022).
9. Liao, Z. *et al.* Fatty acid chain modification enhances the serum stability of antimicrobial peptide B1 and activities against *Staphylococcus aureus* and *Klebsiella pneumoniae*. *Bioorganic Chemistry* **154**, 108015 (2025).
10. Zhao, Y. *et al.* Antimicrobial activity and stability of the d-amino acid substituted derivatives of antimicrobial peptide polybia-MPI. *AMB Express* **6**, 122 (2016).
11. Santos-Júnior, C. D. *et al.* Discovery of antimicrobial peptides in the global microbiome with machine learning. *Cell* **187**, 3761-3778.e16 (2024).
12. Torres, M. D. T. *et al.* Mining human microbiomes reveals an untapped source of peptide antibiotics. *Cell* **187**, 5453-5467.e15 (2024).
13. Ma, Y. *et al.* Identification of antimicrobial peptides from the human gut microbiome using deep learning. *Nature Biotechnology* **40**, 921–931 (2022).
14. Torres, M. D. T., Wan, F. & de la Fuente-Nunez, C. Deep learning reveals antibiotics in the archaeal proteome. *Nature Microbiology* **10**, 2153–2167 (2025).
15. Wan, F., Torres, M. D. T., Peng, J. & de la Fuente-Nunez, C. Deep-learning-enabled antibiotic discovery through molecular de-extinction. *Nature Biomedical Engineering* **8**, 854–871 (2024).
16. Liu, Y. *et al.* A genome and gene catalog of glacier microbiomes. *Nature Biotechnology* **40**, 1341–1348 (2022).
17. Merwin, N. J. *et al.* DeepRiPP integrates multiomics data to automate discovery of novel ribosomally synthesized natural products. *Proceedings of the National Academy of Sciences* **117**, 371–380 (2020).

18. Montalbán-López, M. *et al.* New developments in RiPP discovery, enzymology and engineering. *Natural product reports* **38**, 130 (2020).
19. Agrawal, P., Amir, S., Deepak, Barua, D. & Mohanty, D. RiPPMiner-Genome: A Web Resource for Automated Prediction of Crosslinked Chemical Structures of RiPPs by Genome Mining. *Journal of Molecular Biology* **433**, 166887 (2021).
20. Lei, J. *et al.* The antimicrobial peptides and their potential clinical applications. *American Journal of Translational Research* **11**, 3919–3931 (2019).
21. Elnaggar, A. *et al.* ProtTrans: Toward Understanding the Language of Life Through Self-Supervised Learning. *IEEE Transactions on Pattern Analysis and Machine Intelligence* **44**, 7112–7127 (2022).
22. Hayes, T. *et al.* Simulating 500 million years of evolution with a language model. *Science* **387**, 850–858 (2025).
23. Lin, Z. *et al.* Evolutionary-scale prediction of atomic-level protein structure with a language model. *Science* **379**, 1123–1130 (2023).
24. Abramson, J. *et al.* Accurate structure prediction of biomolecular interactions with AlphaFold 3. *Nature* **630**, 493–500 (2024).

Response to the reviewers

Reviewer #1 (Remarks to the Author):

The authors have addressed most of my prior comments.

Response:

We thank you for the positive evaluation and for recognizing that the previous comments have been addressed.

Reviewer #2 (Remarks to the Author):

It is evident that the authors have undertaken an immense amount of work to address the reviewers' concerns, and their efforts in expanding the experimental validation and refining the genome-quality thresholds have significantly improved the technical rigor of the study. Thank you.

The study approach creates a sequence-function correlation that completely ignores the natural biological context of these peptides. Moreover, it leaves one questioning the rationale for focusing on RiPPs specifically. Theoretically, the authors could have screened the entire sequence space for random peptide libraries and identified active sequence combinations through their model with similar results. By bypassing the very modifications that define RiPPs, the biological relevance of using these specific BGCs as a starting point is significantly diminished. Is this something the authors have ever considered?

The absence of PTMs likely narrowed the chemical space explored and resulted in a convergence of observed modes of action. While the success rate in identifying active peptides (84 out of 100) is technically impressive, the concerns pointed out detract from the study's ability to reveal truly novel or biologically "native" chemical diversity.

Response:

Thanks for the thorough evaluation and constructive comments. We agree that the synthesized peptides do not represent the mature, post-translationally modified forms of RiPP natural products. As correctly noted, many RiPP families, including lassopeptides and lanthipeptides, typically require specific post-translational modifications (PTMs) for their canonical biological functions.

In the present study, our goal was not to reconstruct the full biosynthetic maturation of RiPP natural products, but rather to evaluate the antimicrobial potential encoded in the peptide backbones identified from biosynthetic gene clusters. Our model was trained on primary amino acid sequences, and therefore the experimental validation focused on the intrinsic activity of the predicted peptide scaffolds.

Importantly, RiPP-type BGCs provide a biologically meaningful starting point for peptide discovery, as they encode genetically templated precursor peptides that occur within defined biosynthetic contexts and are associated with dedicated modification enzymes. Even in the

absence of PTMs, the core peptide sequences represent a structured and biologically derived sequence space that serves as a high-quality, non-random library for peptide discovery, which may have contributed to the high success rate (84/100) in identifying active peptides. Moreover, the resulting unmodified scaffolds are synthetically accessible via standard peptide synthesis, making them readily amenable to further optimization and functional characterization. Screening peptides derived from such genomic contexts therefore provides a rational strategy to explore naturally encoded antimicrobial scaffolds.

We agree that the absence of PTMs represents a limitation of the current study. The synthesized peptides should therefore be interpreted as minimal peptide scaffolds rather than mature RiPP natural products. Future work integrating biosynthetic enzymes and modification-aware prediction models will be important to reconstruct the fully modified RiPP molecules and explore the broader chemical diversity encoded in these clusters. We have added the discussion (L. 454-467) in the revised manuscript.

Specific revisions:

We have clarified the limitations of unmodified RiPPs (L. 454-467) in the revised manuscript.

The updated content reads as follows:

L. 454-467: “Although PTMs and stereochemical constraints are often essential to the native bioactivity of RiPPs, the goal of the present study was not to reconstruct the full biosynthetic maturation of RiPP natural products but rather to evaluate the antimicrobial potential encoded in peptide backbones derived from BGCs. Importantly, RiPP-type BGCs provide a biologically meaningful starting point for peptide discovery, as they encode genetically templated precursor peptides that occur within defined biosynthetic contexts and are associated with dedicated modification enzymes. Even in the absence of PTMs, the core peptide sequences represent a structured and biologically derived sequence space that serves as a high-quality, non-random library for peptide discovery, which may have contributed to the high success rate (84/100) in identifying active peptides. Moreover, the resulting unmodified scaffolds are synthetically accessible via standard peptide synthesis, making them readily amenable to further optimization and functional characterization. Screening peptides derived from such genomic contexts therefore provides a rational strategy to explore naturally encoded antimicrobial scaffolds. Nevertheless, the absence of PTMs likely restricts the accessible chemical space of the synthesized peptides and may partly explain the observed convergence in modes of action of cAMPs. Going forward, as curated PTM datasets become more standardized and accessible, incorporating such information with deep learning methods would enable the prediction of enzymatic PTM sites, validation of stereochemistry, and comparisons between modified RiPPs and their unmodified cores.”

Reviewer #3 (Remarks to the Author):

Response:

We thank you and the co-reviewer for your time and contribution to the evaluation and improvement of our manuscript.